# Semi-structured LLM Reasoners Can Be Rigorously Audited

## Abstract

Although Large Language Models (LLMs) have become capable reasoners, the problem of faithfulness persists: their reasoning can contain errors and omissions that are difficult to detect and that may obscure biases in model outputs. To address this issue, we introduce Semi-Structured Reasoning Models (SSRMs), which are trained to produce semi-structured representations of reasoning. SSRMs generate reasoning traces in a *non-executable* Pythonic syntax that names each reasoning step and marks its inputs and outputs. This structure allows SSRM traces to be automatically *audited* to identify reasoning flaws. We evaluate three types of audits: hand-crafted *structured reasoning audits*, written in a domain-specific language (DSL) implemented in Python; LLM-generated *structured reasoning audits*; and learned *typicality audits*, which apply probabilistic models over reasoning traces. We show that all of these methods can be used to effectively flag probable reasoning errors. Importantly, the auditability of SSRMs does not appear to compromise overall accuracy: in evaluation on twelve benchmarks and two model families, SSRMs demonstrate strong performance and generalizability relative to other models of comparable size. We provide our code at Anonymous Github Link.

## 1 Introduction

Large Language Models (LLMs) often benefit from reasoning techniques such as short Chain-of-Thought (CoT) prompting (Wei et al., 2022) or long CoT reasoning (Chen et al., 2025a; Wang et al., 2025; Wang, 2025). Yet in many applications, LLMs may generate superficially plausible but incorrect reasoning that obscures biases in the output (Turpin et al., 2024); more generally, reasoning traces are not causally related to the final output (Bao et al., 2024). This problem of "unfaithful"LLM reasoning has been extensively investigated in short CoT settings (Lanham et al., 2023; Bentham et al., 2024; Parcalabescu & Frank, 2024), and is likely to be more problematic in long CoT reasoning.

As a concrete step toward demystifying reasoning LLMs and improving their reliability, we present methods for *rigorously checking LLM reasoning on specific tasks*. To illustrate and motivate this, consider the simplified medical question-answering (QA) task in Figure 1, adapted from the Med-CalcBench (Khandekar et al., 2024). The "flawed" reasoning trace appears superficially plausible but is incomplete compared to the "ideal" trace: it contains one obvious omission, one subtler error, and one issue where the LLM fails to explicitly check the compatibility of the units for an extracted value. Although none of these affect the final answer in this example, such flaws are undesirable in consequential tasks. Human experts performing similar tasks are often expected to carefully follow explicit instructions—variously called rubrics, cookbooks, or policies depending on the domain—to ensure that reasoning is complete and decisions are made consistently. This observation motivates the central research question: *can we detect when an LLM deviates from a desired reasoning strategy?*

Since analyzing arbitrary reasoning traces is difficult, we begin by training an LLM to generate *semi-structured* reasoning traces, as shown in Figure 2 (Top). Following prior work (Cohen & Cohen, 2024), we adopt a Pythonic syntax that labels different types of reasoning steps using a restricted, task-specific vocabulary and specifies the *inputs and outputs* of each step, *without requiring the steps to be executable*. Because the steps can perform arbitrary computations and consume or produce arbitrary strings of text, this semi-structured format is highly general. In this paper we provide new evidence for this generality by showing that training models to generate *semi-structured* traces achieves performance comparable to similarly trained free-form reasoning models and other baselines.

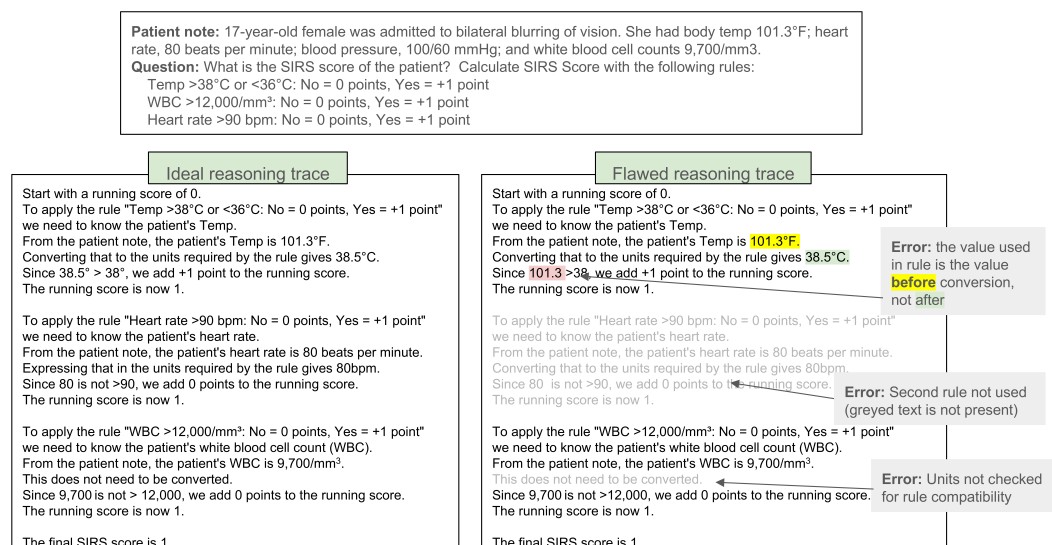

Figure 1: Overview of the problem addressed. Top: a question that requires the LLM to extract information and apply reasoning to answer correctly. Bottom left: a desired reasoinng trace. Bottom right: a flawed reasoning trace. The flawed trace differs from the desired one in three ways: (1) the incorrect patient measurement is used to determine applicability of the first rule; (2) the second rule is skipped; (3) the units associated with a patient measurement are not explicitly checked against those required by the third rule. In this example, none of these reasoning flaws affects the final answer, so this flawed reasoning trace will be reinforced during reinforcement learning with an outcome reward.

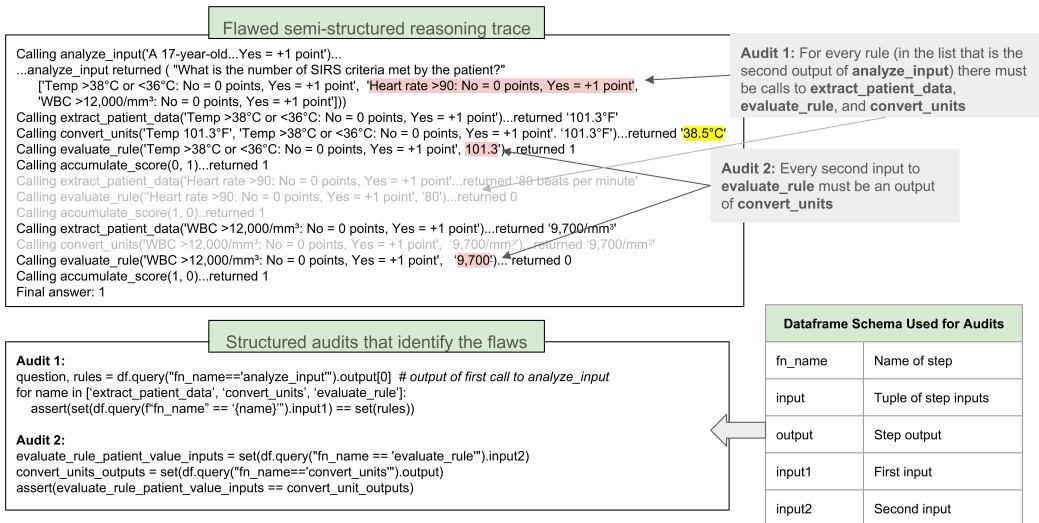

Figure 2: Overview of our approach. An LLM is trained to generate a *semi-structured trace* comprising a function name, its inputs, and its outputs for each reasoning step. Two plausible constraints on this semi-structured trace are also shown, given in natural language (gray boxes) and as executable tests (bottom left). The executable tests are *reasoning trace audits*, and in this case are hand-written. We also explore *typicality audits*, which are learned from a corpus of reasoning traces.

We further show that semi-structured reasoning *facilitates the scalable detection of reasoning flaws*. For example, in Figure 2, one can observe that a desired rule was skipped by comparing the number of `evaluate_rule` steps with the length of the rule list returned by `analyze_input`. We refer to such

checks as *reasoning audits*. Figure 2 also provides natural-language descriptions of two audits alongside their corresponding *structured reasoning audits*. Our results indicate that these manually implemented audits can identify potential reasoning flaws and flag outcomes that are likely incorrect.

Our DSL for structured queries uses trace that has been encoded as a Pandas DataFrame, and audits also look like Python unit tests—two widely-used programming constructs. Because of these design choices, we show that *structured reasoning audits are also easily generated automatically by modern LLMs given minimal guidance*, substantially reducing the cost of auditing reasoning in a new domain.

Beyond enabling structured reasoning queries, *semi-structured reasoning facilitates additional forms of analysis*. A recurring question in the literature (Kambhampati et al., 2025) is whether reasoning LLMs generate novel "reasoning patterns" or simply reproduce patterns that are seen during training. This issue is difficult to address without a formal definition of "reasoning patterns." In this work, we explore certain definitions of a "reasoning pattern" for semi-structured reasoners and use it to build *probabilistic models of reasoning patterns for specific tasks*. We evaluate the hypothesis that model accuracy correlates with the probability assigned to its reasoning patterns. By analogy with structured audits, we term this a *typicality audit*, and show that they can also identify potential reasoning errors.

In summary, this paper makes the following contributions:

- We introduce two-stage training recipes for SSRMs that produce semi-structured reasoning traces.
- We illustrate that both manually-generated and LLM-generated structured audits can effectively reveal potential reasoning flaws, and that failing certain audits increases the probability of error.
- We show that typicality audits can reveal common reasoning patterns linked to better outcomes.
- We demonstrate that auditability comes without cost in generalization performance, as SSRMs achieve results comparable to similarly trained unstructured reasoning models and other baselines.

## 2 RELATED WORK

**Faithfulness and Process Models.** CoT prompting has been shown to sometimes produce predictions that preserve underlying LLM biases, accompanied by explanations that obscure those biases (Turpin et al., 2024). This observation has motivated extensive research on explanation faithfulness in CoT prompting Jacovi & Goldberg (2020); Turpin et al. (2024); Lanham et al. (2023); Bao et al. (2024). Nevertheless, defining and measuring faithfulness remains challenging, with some prior studies advocating quantitative approaches that assess mechanistic influence in neural networks through numerical metrics (Parcalabescu & Frank, 2024; Bentham et al., 2024; Chen et al., 2025b). In this work, we propose *reasoning audits* as a concrete and testable alternative to measuring faithfulness.

Other studies have proposed methods verifying reasoning chains using *process reward models* (Paul et al., 2024; Sun et al., 2024b) and *step reward models* Viteri et al. (2024); Wang et al. (2023); Saparov & He (2022); Lai et al. (2024). However, these reward models are typically tailored to specific domains—such as mathematics (Paul et al., 2024; Sun et al., 2024b) or theorem-proving (Saparov & He, 2022; Lai et al., 2024)—and often rely on Monte Carlo Tree Search (Kocsis & Szepesvári, 2006) to explore and evaluate multiple candidate reasoning chains, a computationally expensive procedure. While our work is largely orthogonal, the symbolic and statistical audits we propose could provide complementary signals for future reward-model training. In particular, the statistical audits we proposed refine the notion of reasoning patterns, which have previously been identified either through task-specific analyses (Zhang et al., 2025) or via LLM pipeline methods (Zhou et al., 2025).

**Semi-structured LLM Reasoning.** Various prompting strategies—such as CoT (Wei et al., 2022), Tree-of-Thought (ToT) (Yao et al., 2023), Chain-of-Code (CoC) (Li et al., 2023), and Program-of-Thought (PoT) (Chen et al., 2022)—have been widely employed to enhance the reasoning capabilities of LLMs. More recently, research has shifted from prompting toward inference-time scaling by incorporating search algorithms, particularly tree-based search (including Monte Carlo Tree Search variants) and beam search, into the sampling process (Feng et al., 2023; Trinh et al., 2024; Xin et al., 2024; Kocsis & Szepesvári, 2006); by ensembling multiple reasoning trajectories through self-consistency (Wang et al., 2022; Huang et al., 2025; Aggarwal et al., 2023); and by applying reinforcement learning (RL) to extend the length of reasoning (OpenAI, 2024; Shao et al., 2024; Guo et al., 2025; Qwen, 2024; Kumar et al., 2024; Yang et al., 2025). Despite these, LLMs frequently produce reasoning traces that appear plausible yet incorrect, and such errors can be difficult to detect.

Previous studies have proposed that faithfulness can be improved by using a code-like format for LLM outputs. Prior work assumes this format is either fully executable Python programs (Chen et al., 2022; Gao et al., 2023; Lyu et al., 2023; Paranjape et al., 2023) or partially executable pseudocode (Li et al., 2023; Weir et al., 2024; Chae et al., 2024). While enabling the use of Python as a tool often improves performance, the reasoning process used to generate the pseudocode remains obscured. These works have also argued (sometimes implicitly) that faithfulness is qualitatively improved with code-based outputs. In contrast, we pursue the more concrete goal of auditing the reasoning process.

We build most on the reasoning-chain syntax used in Program Trace Prompting (PTP) (Cohen & Cohen, 2024). While PTP uses few-shot prompting to extrapolate "partial programs" and sample traces for novel inputs, SSRMs achieve strong performance without task-specific few-shot prompts.

## 3 TRAINING METHODS

We use a two-stage training recipe for a Semi-Structured Reasoning Model (SSRM). The first stage performs SFT to teach the model to produce the semi-structured reasoning traces, while the second stage uses reinforcement learning with verifiable rewards (RLVR) to enhance the reasoning ability.

**Supervised Fine-Tuning.** To collect SFT data for semi-structured reasoning, we follow the PTP approach (Cohen & Cohen, 2024). We generate semi-structured reasoning traces with PTP using both Claude Sonnet 3.5 and 3.7 (Anthropic, 2024; 2025) on a subset of BBH tasks (Suzgun et al., 2022) as well as subsets of the training data from GSM8K (Cobbe et al., 2021), MATH500 (Lightman et al., 2023), and MedCalcBenchV2 (please see Section 4). Only traces that produce a correct final answer are retained. To verify correctness, we extract answers from the answer tags and evaluate their accuracy. We also perform a simple formatting check to remove samples whose partial programs or traces cannot be parsed. For the final dataset, we apply downsampling to balance the number of samples across tasks. The distribution of the resulting SFT data is provided in Appendix E.1 Table 9.

**Chain-of-Thought Baseline.** To establish a fair baseline for comparison, we construct a standard CoT dataset. We generate CoT traces on BBH using the original few-shot prompts applied to Claude Sonnet 3.5, and augment the training data with ground-truth CoT solutions from GSM8K, MATH500, and MedCalcBenchV2, for the same problem instances used in the semi-structured SFT training data.

**Training Template.** We structure each example using a consistent markup format. In the semi-structured setting, partial programs are wrapped in `<partial_program>` tags, reasoning traces in `<program_trace>` tags, both enclosed within a `<think>` tag. The final answer is placed inside the `<answer>` tag for easy parsing. For the CoT baseline, only `<think>` and `<answer>` tags are used.

**RLVR Dataset.** In the second stage, we enhance the SFT model with RLVR. We construct the RLVR dataset by sampling eight responses per problem from the English subset of DAPO-Math-17K (Yu et al., 2025), using the SFT checkpoint. We randomly discard half of the samples with pass rates of either zero or one. We further include a held-out subset of MedCalcBenchV2 excluded from SFT.

**Reward Design.** We adopt a rule-based reward combining outcome accuracy and structural validity. Outcome accuracy measures the correctness of the final answer, while format rewards are assigned if the reasoning trace conforms to the semi-structured or CoT format, evaluated via regular expressions.

**RL Algorithm.** We optimize with the Group Relative Policy Optimization (GRPO) (Shao et al., 2024), which estimates token-level advantages without requiring a critic. For a specific question-answer pair $(q, a)$, the policy model first samples a group of $G$ individual responses $\{\mathbf{o}_i\}_{i=1}^{G}$. Subsequently, the advantage of the $i$-th response is calculated as $A_{i,t} = \frac{r_i - \text{mean}\left(\{R_i\}_{i=1}^{G}\right)}{\text{std}\left(\{R_i\}_{i=1}^{G}\right)}$. And the training objective is

$$\mathcal{J}_{\text{GRPO}}(\theta) = \mathbb{E}_{(q,a)\sim\mathcal{D}, \{\mathbf{o}_i\}_{i=1}^{G}\sim\pi_{\theta_{\text{old}}}(\cdot|q)}$$

$$\left[ \frac{1}{G}\sum_{i=1}^{G}\frac{1}{|\mathbf{o}_i|}\sum_{t=1}^{|\mathbf{o}_i|}\left(\min\left(r_{i,t}(\theta)A_{i,t}, \text{clip}\left(r_{i,t}(\theta), 1-\varepsilon, 1+\varepsilon\right)A_{i,t}\right) - \beta D_{\text{KL}}\left(\pi_\theta \| \pi_{\text{ref}}\right)\right)\right],$$

where $r_{i,t}(\theta) = \frac{\pi_\theta\left(\mathbf{o}_{i,t} \mid q, \mathbf{o}_{i,<t}\right)}{\pi_{\theta_{\text{old}}}\left(\mathbf{o}_{i,t} \mid q, \mathbf{o}_{i,<t}\right)}$

$$\tag{1}$$

Differ from standard GRPO, we adopt fully on-policy training and token-level loss (Yu et al., 2025).

# 4 EXPERIMENTS

In this section, we present a series of experiments conducted across diverse benchmarks—including mathematics, medical, and health domains—covering both in-domain datasets and those outside the training mixture. We also compare SSRMs to strong prompted baselines. Our goal is to address three key questions: (1) Can the reasoning traces of SSRMs be audited, either through structured queries or statistical methods? (2) Can prompted models be audited in a similar manner? (3) Is semi-structured reasoning more difficult to learn? Detailed setups and dataset descriptions are listed in Appendix F.

**Experimental Setup.** We use Qwen2.5-7B (Yang et al., 2024) as the base model for SSRM and conduct auditability analysis on its generated semi-structured reasoning traces. To further validate the performance, we also train an SSRM based on Llama3.1-8B (Grattafiori et al., 2024). All models are trained using verl (Sheng et al., 2024) on 8 H100 GPUs, with evaluations conducted on 1 H100.

In addition to similarly trained unstructured baselines, we compare SSRMs against baselines of comparable size. Non-reasoning baselines include Llama3.1-8B-Instruct (Grattafiori et al., 2024), Medical Llama (ContactDoctor, 2024) (fine-tuned for biomedical knowledge), and the Qwen series (Yang et al., 2024). Reasoning baselines include the DeepSeek-Distilled series (Guo et al., 2025). For prompted baselines, we evaluate Claude Sonnet 3.5 (Anthropic, 2024) and Qwen2.5-7B-Instruct.

We use greedy decoding and report accuracy for all tasks, except for AIME24, where we sample 32 responses and report Pass@1 with a temperature of 0.7. The maximum generation length is set to 32,768 tokens. All tasks are evaluated in a zero-shot setting, except for prompted baselines, which use two-shot prompts. Reasoning baselines follow the recommended setting (temperature 0.6, top-$p$ 0.95) (Guo et al., 2025). For Qwen2.5-7B, we omit the chat template following Liu et al. (2025).

**Primary Evaluation: MedCalcBenchV2.** Our primary evaluation benchmark is MedCalcBenchV2, a cleaned version of MedCalcBench (Khandekar et al., 2024) (See Appendix F.1). MedCalcBenchV2 measures an LLM's ability to extract information from clinical text (*patient note*) and perform calculations using either explicit rules or formulas provided in the prompt. We observe that rule-based tasks are substantially more challenging than formula-based tasks. Errors in formula-based problems primarily arise from computation or extraction mistakes, whereas errors in rule-based problems more often involve failures to follow explicit instructions, consistent with prior findings on rule-following tasks (Sun et al., 2024a). To account for this discrepancy, we treat the two categories as two distinct tasks: MedCalcV2 Rules and MedCalcV2 Formulas. Evaluation follows the original MedCalcBench criteria, which allow small numeric deviations and employ rule-based checks for date-based answers.

**Domain-Specific Language for structured audits.** Our DSL for structured audits looks like Python unit tests: they are class methods, can be called without arguments, and contain assertion statements invoked by the class method `self.assertTrue`. An audit fails if it raises an exception or if an `assertTrue` call does not hold. The method can access a Pandas DataFrame `self.df` that represents the semi-structured trace, and assertions usually operate on this data structure using Pandas operations.

**Additional Evaluation.** To evaluate the generalizability of the SSRMs beyond in-domain data, we conduct additional experiments on a range of benchmarks: general reasoning (GPQA-Diamond (Rein et al., 2024)), mathematical reasoning (AIME24), commonsense reasoning (CommonsenseQA (Talmor et al., 2018)), truthfulness (TruthfulQA (Lin et al., 2021)), as well as several medical and health-related tasks, namely MedQA (Jin et al., 2020), the biology and health subsets of MMLU-Pro (Wang et al., 2024), and PubMedQA (Jin et al., 2019), which we convert to multiple-choice.

## 4.1 EXPERIMENTAL RESULTS

**Both hand-crafted and LLM-generated structured audits are effective for auditing semi-structured reasoning traces generated by SSRMs.** To validate that semi-structured reasoning can be systematically audited, we first apply hand-crafted audits for the two MedCalcV2 tasks based on the analysis of the training examples. Table 1 reports results for all individual audits that are applied with sufficient frequency[1] and are sufficiently discriminative—specifically, audits that succeed at least 5% of the time and fail at least 5% of the time. The second audit for MedCalcV2 Formulas (e.g., "math is correct") uses Python's `eval` function; whereas all other audits inspect only trace structure.

---

[1] Audits may not be applied to all traces—for example, one cannot confirm that number of rules evaluated is the same as the number of rules extracted if rule extraction fails to produce a legal output.

Table 1: Hand-crafted structured audits for Qwen SSRM generated semi-structured traces on two MedCalcV2 tasks. For each, we report the failure rate, the outcome accuracy conditioned on audit failing or passing, the accuracy difference ($\Delta$) between passing and failing cases, and the $p$-value for testing $\Delta \neq 0$. One star (*) for statistical significance at $p < 0.1$ and two stars (**) for $p < 0.05$.

| | | — accuracy and difference — | | | | |
| | %Failed | Failing | Passing | $\Delta$ | $p$-val | description of audit |
|---|---|---|---|---|---|---|
| MedCalcV2 Formulas | 22.0 | 0.77 | 0.86 | 0.09 | | `solve_formula` output is formatted well |
| | 49.0 | 0.84 | 0.83 | -0.01 | | `solve_formula` math is correct[math] |
| MedCalcV2 Rules | 13.2 | 0.22 | 0.46 | 0.24 | ** | one `get_data` step per extracted rule |
| | 13.4 | 0.22 | 0.47 | 0.25 | ** | `get_data` called on all rules |
| | 14.0 | 0.21 | 0.47 | 0.26 | ** | one `eval_rule` step per rule |
| | 20.3 | 0.26 | 0.48 | 0.22 | ** | all rule outputs summed correctly |

Table 2: LLM-generated structured audits on the same set of Qwen SSRM traces for MedCalcV2.

| | | — accuracy and difference — | | | | |
| | %Failed | Failing | Passing | $\Delta$ | $p$-val | description of audit |
|---|---|---|---|---|---|---|
| MedCalcV2 Formulas | 5.7 | 0.76 | 0.84 | 0.08 | | step 4 output feeds into step 5 input |
| | 6.4 | 0.45 | 0.86 | 0.41 | ** | step 3 output feeds into step 4 input |
| | 8.3 | 0.62 | 0.86 | 0.24 | | step 2 output feeds into step 3 input |
| | 9.3 | 0.82 | 0.84 | 0.02 | | `convert_units` called once per datapoint |
| | 10.2 | 0.81 | 0.84 | 0.03 | | `convert_units` receives formula as first input |
| | 12.6 | 0.80 | 0.84 | 0.05 | | `convert_units` correct second input |
| | 14.4 | 0.37 | 0.92 | 0.55 | ** | `get_data` receives formula from `analyze_input` |
| MedCalcV2 Rules | 13.4 | 0.22 | 0.47 | 0.25 | ** | `get_data` called for each rule |
| | 13.9 | 0.21 | 0.47 | 0.26 | ** | consistent rules across `get_data` steps |

As suggested in Figure 2, reasoning flaws do not always yield incorrect outcomes. In Table 1, for each audit $a$, we present test accuracy when $a$ fails ("Failing" column), when $a$ passes ("Passing" column), the accuracy difference $\Delta$, and the statistical significance of the difference being non-zero.

The results suggest that reasoning errors are more frequent in MedCalcV2 Rules than in Formulas. While math errors in Formulas occur frequently, they do not correlate with outcome errors.[2] In contrast, reasoning errors in Rules are common and associated with substantial accuracy losses. The most common failure is mis-summing rule contributions, followed by skipping a rule. Other failing audits indicate mismatches between the counts of patient data extraction and rule application steps.

Because manually generating audits is expensive, we also explore automatic generation of structured audits using LLMs. We manually write audits for three additional tasks from BBH, and use those as few-shot examples to prompt Claude Sonnet 4.0 to output structured audits given a set of three correct sample traces. The results, shown in Table 2, show that LLM-generated structured audits are comparably useful to hand-crafted ones. (For more results and details, please check Appendix F.4)

**Typicality audits are also applicable for auditing semi-structured reasoning traces generated by SSRMs.** Typicality audits provide a complementary use of the semi-structured format by analyzing *abstract versions of reasoning processes*, aka "reasoning patterns" (Zhang et al., 2025). Prior work has conjectured that LLMs predominantly reproduce "reasoning patterns" observed in the training data and struggle to generate novel sequences—i.e., LLM reasoning often relies on retrieving previously seen reasoning examples (Kambhampati et al., 2025). If this holds, reasoning within a given task should exhibit regularity, thereby enabling statistical analyses to flag outlier traces as potential errors.

In past work, "reasoning patterns" are typically identified heuristically or by LLMs (Zhang et al., 2025; Zhou et al., 2025). Here we define "reasoning patterns" as the sequence of step names. For example, in Figure 2, the pattern is "`analyze_input extract_patient_data convert_units evaluate_rule accumulate_score extract_patient_data evaluate_rule accumulate_score`". We then construct a probabilistic model $M$ over these sequences, treating them as language tokens. This formulation yields a precise version of the conjecture above: *LLM correctness is positively correlated with the*

---

[2]MedCalcV2 numerical answers are soft-matched to the target, whereas the implemented audits check exact equivalence before and after simplification.

*probability of the required reasoning pattern under $M$.* To test this, we compute the correlation between outcome correctness and the probability of the reasoning pattern generated by the SSRMs.

We consider the following types of *reason pattern typicality* models $M$: a *unigram* language model smoothed with a Dirichlet prior (referred to as *multinomial* in the tables below); *bigram* and *trigram* models, implemented simply by extending the base vocabulary to consider all $n$-grams of step names for $n = 2, 3$; an HMM with three hidden states over trigrams, denoted *HMM(3,3)* in the table; and a final model, *HMM\**, in which we perform a grid search over different $n$-gram sizes and numbers of hidden states, selecting the configuration that optimizes the BIC criterion (Please see Appendix F.2).

Table 3 summarizes the results obtained by fitting these models to the test data.[3] We use Kendall's $\tau$ to measure correlation because it makes no parametric assumptions and observe only moderate correlations ranging from 0.08 to 0.26. As another way of testing if highly typical reasoning patterns correspond to higher accuracies than atypical ones, we sort all test predictions by pattern probability and compare the accuracy of the least probable third with that of the most probable third, excluding the middle third. Using this method, we observe significant differences in many models. For example, under the *HMM(3,3)* model, the accuracy difference ($\Delta$) between the least and most probable tertiles is approximately 25% for both MedCalcV2 tasks. These results suggest that *a typicality model $M$ can approximate the behavior of a structured audit* by: (a) sorting predictions by typicality, (b) splitting predictions into quantiles, and (c) interpreting the top quantile as audit-passed cases, the bottom quantile as audit-failed cases, and any other intermediate quantiles as unevaluable.

Table 3: The results prove that atypical reasoning pattern in the MedCalcV2 tasks are more likely to result in errors. We evaluate several typicality/probability models, all of which correlate with correctness, though the correlation is weaker on MedCalcV2 Rules. In additional to Kendall's $\tau$ for correlation, we also partition the test data into three equal groups by probability and report accuracy in the lowest and highest tertiles, the accuracy difference ($\Delta$), and the $p$-value of this difference.

| MedCalcV2 Formulas | | – accuracy and difference – | | |
|---|---|---|---|---|
| | $\tau$ | Tertile 1 | Tertile 3 | $\Delta$ | $p$-val |
| multinomial | 0.25 | 0.72 | 0.95 | 0.22 | * |
| bigram | 0.25 | 0.72 | 0.95 | 0.23 | * |
| trigram | 0.26 | 0.72 | 0.95 | 0.23 | * |
| HMM(3,3) | 0.26 | 0.72 | 0.97 | **0.25** | ** |
| HMM* | 0.21 | 0.74 | 0.97 | 0.07 | |

| MedCalcV2 Rules | | – accuracy and difference – | | |
|---|---|---|---|---|
| | $\tau$ | Tertile 1 | Tertile 3 | $\Delta$ | $p$-val |
| multinomial | 0.17 | 0.32 | 0.57 | **0.25** | ** |
| bigram | 0.17 | 0.32 | 0.57 | **0.25** | ** |
| trigram | 0.17 | 0.32 | 0.57 | **0.25** | ** |
| HMM(3,3) | 0.17 | 0.32 | 0.57 | **0.25** | ** |
| HMM* | 0.08 | 0.43 | 0.52 | 0.09 | |

Following this approach, the typicality audits function as an *abstaining classifier* Pietraszek (2005) for evaluating outcome correctness, abstaining specifically on intermediate scores. The abstention rate can be adjusted by splitting the predictions into different numbers of quantiles: for instance, dividing predictions into two groups yields no abstentions, whereas dividing them into eight groups results in abstention for the middle six octiles—i.e., $\frac{6}{8} = \frac{3}{4}$ of the time. As shown in Figure 3, higher abstention rates are associated with larger accuracy difference ($\Delta$) across both MedCalcV2 tasks.

Given the effectiveness of both structured and typicality audits in identifying potential reasoning errors, a straightforward extension is to apply at inference time. For example, when combining typicality audits with self-consistency, reasoning traces in the lowest tertile can be resampled more extensively, whereas those in the highest tertile—more likely to be correct— might require no additional sampling. This strategy can help concentrate the sampling budget on the most error-prone cases. We evaluate this approach on the MedCalcV2 Rules tasks and report the results in Appendix F.7. Our findings show that audit-guided self-consistency reduces computational cost while maintaining compara-

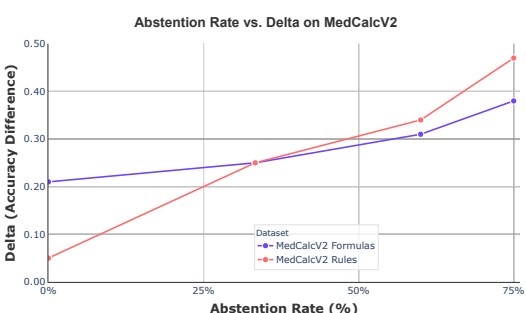

Figure 3: Abstention Rate vs. $\Delta$ (accuracy difference between the highest- and lowest- probability quantiles) on MedCalcV2 under a typicality audit.

ble or slightly improved performance relative to vanilla self-consistency with a fixed sampling budget. However, we do not observe significant improvements over greedy decoding with a single sample.

---

[3]Note that the correctness label is not used in this step. Moreover, the distribution of reasoning patterns in the training data may differ for tasks with ground-truth reasoning chains.

Table 4: Results of applying structured audits to Claude Sonnet 3.5 with semi-structured prompting on both MedCalcV2 Formulas and Rules. Overall, the results resemble those of SSRM, although the prompted MedCalcV2 Formulas system does have some rarely-failing audits that impact accuracy.

| | %Failed | — accuracy and difference — | | | $p$-val | description of audit |
| | | Failing | Passing | Δ | | |
|---|---|---|---|---|---|---|
| MedCalcV2 Formulas | 1.7 | 0.000 | 0.662 | 0.662 | | one `get_data` step |
| | 2.1 | 0.000 | 0.664 | 0.664 | * | one `insert_variables` step |
| Claude Sonnet 3.5 | 3.8 | 0.091 | 0.673 | 0.582 | ** | `solve_formula` output is a number[math] |
| (65.1% acc) | 9.2 | 0.593 | 0.657 | 0.064 | | `solve_formula` output is formatted correctly |
| | 47.3 | 0.667 | 0.636 | -0.030 | | `solve_formula` math is correct[math] |
| MedCalcV2 Rules | 5.8 | 0.182 | 0.399 | 0.218 | | `analyze_input` returns correct # values |
| | 14.7 | 0.196 | 0.420 | 0.223 | ** | one `convert_units` step per rule |
| Claude Sonnet 3.5 | 14.7 | 0.196 | 0.420 | 0.223 | ** | one `get_data` step per rule |
| (38.7% acc) | 15.8 | 0.183 | 0.425 | 0.242 | ** | one `evaluate_rule` step per rule |
| | 17.1 | 0.169 | 0.432 | 0.263 | ** | one `accumulate_score` step per rule |

We hypothesize that traces flagged as incorrect by audits may correspond to problems that the model struggles to solve even with additional sampling budget. We leave this to future work.

**Both structured and typicality audits can be applied to semi-structured traces from few-shot prompted models.** Table 4 presents the results of structured audits applied to few-shot prompted Claude Sonnet 3.5 on both Med-CalcV2 tasks, while Table 5 shows typicality audit results for the same model (limited to three representative typicality models for brevity). Overall, Claude Sonnet 3.5 behaves similarly to SSRMs under these audits, except for the typicality audit for Rules, where more typical reasoning traces exhibit *higher* error rates on average (although not significantly so). This may be attributed to the high error rate of the prompted model itself: even typical reasoning processes often lead to errors. In Appendix F.4, we present results for Qwen2.5-7B-Instruct— a weaker prompted model and the instruction-tuned version of the Qwen2.5-7B base used as the backbone for SSRMs—as well as LLM-generated structured audit results on prompted Claude.

Table 5: Results of applying typicality audits to semi-structured reasoning traces from few-shot prompted Claude Sonnet 3.5 on both MedCalcV2 Formulas and Rules. Overall, the results indicate that the hypothesis—that atypical reasoning patterns correspond to higher error rates—holds for MedCalcV2 Formulas but not for the noisier Rules.

| | $\tau$ | — accuracy and difference — | | | $p$-val |
| | | Tertile 1 | Tertile 3 | Δ | |
|---|---|---|---|---|---|
| MedCalcV2 Formulas | | | | | |
| Claude Sonnet 3.5 (65.1% acc) | | | | | |
| trigram | 0.13 | 0.56 | 0.67 | 0.11 | |
| HMM(3,3) | 0.21 | 0.56 | 0.70 | 0.14 | |
| HMM* | 0.30 | 0.54 | 0.87 | 0.33 | ** |
| MedCalcV2 Rules | | | | | |
| Claude Sonnet 3.5 (38.7% acc) | | | | | |
| trigram | -0.06 | 0.47 | 0.33 | -0.14 | |
| HMM(3,3) | -0.06 | 0.46 | 0.32 | -0.14 | |
| HMM* | -0.05 | 0.43 | 0.33 | 0.00 | |

**Semi-structured reasoning is learnable and achieves performance and generalization on par with unstructured reasoning.** We consider two different model families for training SSRMs: a stronger one based on Qwen2.5-7B and a weaker one based on Llama3.1-8B. Our trained Qwen SSRM achieves strong results, as shown in Table 6. On average, *it outperforms the unstructured reasoning baseline trained with the same procedure*. On the two challenging MedCalcV2 tasks, it exceeds six other strong baselines of comparable size. Within the training mixture, it outperforms the best baseline by

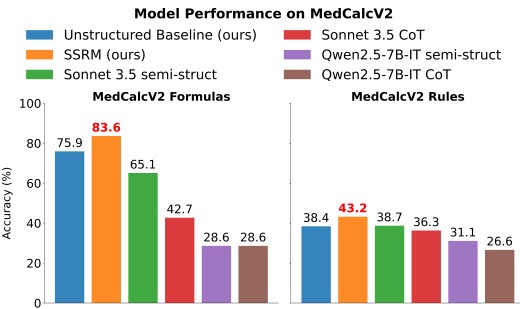

Figure 4: Comparison of SSRM against similarly trained and prompted baselines on MedCalcV2.

nearly ten points. Moreover, it generalizes effectively to tasks outside the training mixture: while it does not match the top math-specialized reasoning models, it outperforms all non-reasoning baselines. On a range of medical QA benchmarks, it achieves performance comparable to reasoning models, lagging only slightly behind BioMedical-Llama-3-8B, a specialized model for biomedical knowledge. By contrast, although the Llama SSRM is based on a weaker backbone, it delivers performance comparable to similarly trained unstructured reasoning baselines on both in-domain and out-of-domain benchmarks, further supporting that semi-structured reasoning does not compromise performance.

Table 6: Our models are initialized from Qwen2.5-7B and Llama3.1-8B, trained with SFT followed by RLVR on a mix of MedCalcV2 and other math tasks. Underlined results indicate the best performance among our comparably trained models; starred results denote best among non-reasoning models; and bold results are best overall. On average, SSRM outperforms the unstructured CoT format and six strong, comparably sized baseline models.[4] We sample 32 times and report Pass@1 for AIME24.

| | SSRM from Qwen2.5-7B (ours) | | | | | SSRM from Llama3.1-8B (ours) | | | | Instr/reasoning LLMs (Qwen2.5-7B) | | | Instr/reasoning LLMs (Llama3/3.1-8B) | | |
|---|---|---|---|---|---|---|---|---|---|---|---|---|---|---|---|
| | Base | unstr. +SFT | unstr. ++RL | semi-str. +SFT | semi-str. ++RL | unstr. +SFT | unstr. ++RL | semi-str. +SFT | semi-str. ++RL | Instr | *OpR1 | *DSeek | BioL | Instr | *DSeek |
| MedCalcV2 Formulas | 3.0 | 52.4 | 75.9 | 63.3 | *__83.6__ | 48.7 | 58.9 | 56.9 | 75.0 | 56.4 | 44.8 | 36.9 | 12.6 | 10.0 | 29.7 |
| MedCalcV2 Rules | 0.0 | 27.4 | 38.4 | 38.9 | *__43.2__ | 27.9 | 28.4 | 20.8 | 36.3 | 32.1 | 22.6 | 14.2 | 16.6 | 9.5 | 9.2 |
| GSM8k | 85.4 | 74.4 | 90.5 | 76.6 | *90.9 | 42.3 | 43.8 | 57.4 | 36.2 | *90.9 | **94.8** | 89.2 | 51.9 | 81.1 | 75.7 |
| MATH500 | 69.2 | 44.6 | 77.0 | 45.4 | 75.2 | 15.6 | 15 | 22.6 | 12.4 | *78.8 | 91.0 | 94.0 | 17.4 | 46.2 | 87.2 |
| *train mix avg* | 39.4 | 49.7 | 70.5 | 56.1 | *73.2 | 33.6 | 36.5 | 39.4 | 40 | 64.6 | 63.3 | 58.6 | 24.6 | 36.7 | 50.5 |
| AIME24 | 9.1 | 1.1 | 12.1 | 3.7 | *12.4 | 0.3 | 0.5 | 0.2 | 0.9 | 11.8 | 45.3 | **53.4** | 0.1 | 2.1 | 45.2 |
| GPQA-D | 31.8 | 31.8 | *38.4 | 30.3 | 34.3 | 25.8 | 33.8 | 22.2 | 26.3 | 32.8 | 41.4 | **50.5** | 26.8 | 31.8 | 43.9 |
| TruthfulQA | 49.7 | 57.3 | 56.3 | 41.1 | 54.3 | 41.6 | 39.8 | 31.3 | 32.9 | *55.6 | 42.6 | 47.5 | 53.0 | 54.3 | 52.6 |
| CommonsenseQA | 70.5 | 70.1 | 72.8 | 70.8 | *75.7 | 52.7 | 52.4 | 58.6 | 60.4 | 66.8 | 54.0 | 52.3 | 39.3 | 50.4 | 63.1 |
| MedQA | 57.4 | 62.4 | 62.0 | 55.9 | 61.4 | 55.4 | 57 | 50.4 | 53 | *62.8 | 31.1 | 36.4 | **76.9** | 68.9 | 58.1 |
| MMLU Pro Bio | 64.6 | 68.6 | 71.8 | 59.3 | 69.9 | 58 | 58.4 | 55.4 | 59.3 | *73.5 | 50.9 | 66.7 | 64.6 | 67.8 | 73.1 |
| MMLU Pro Health | 42.1 | 53.2 | 53.1 | 40.5 | 51.7 | 40.1 | 41.6 | 40.1 | 39.7 | *54.8 | 22.0 | 33.4 | 53.1 | 58.3 | 46.5 |
| PubmedQA | 66.3 | 73.4 | 71.4 | 70.2 | *76.2 | 73.9 | 75.5 | 68.2 | 73.4 | 73.5 | 73.3 | 72.7 | **77.1** | 75.6 | 73.8 |
| *med/health avg* | 57.6 | 64.4 | 64.6 | 56.5 | 64.8 | 56.9 | 58.1 | 53.5 | 56.4 | 66.2 | 44.3 | 52.3 | *67.9 | 67.7 | 62.9 |
| *overall avg* | 45.3 | 51.3 | 60.8 | 50.2 | *61.7 | 39.7 | 41.7 | 40.3 | 42 | 58.0 | 52.1 | 54.3 | 39.5 | 45.6 | 54.6 |

[4] All accuracies are percentages. "Instr" models are instruction-trained, "DSeek" are distilled from DeepSeek-R1, and OpR1 is OpenR1-Qwen-7B. BioL is Bio-Medical-Llama-3-8B.

In Figure 4, we show that Qwen SSRM not only outperforms the Claude Sonnet 3.5, which is used to seed the SFT training data, but also significantly outperforms the Qwen instruction-tuned variant. For Claude Sonnet 3.5 and Qwen2.5-7B-Instruct, we employ two-shot prompting, using two fixed demonstrations across all MedCalc "calculators".[5] For each prompted model, we evaluate two prompt variants: one with unstructured free-form CoT prompts and one with the semi-structured format.

We also analyze the token usage of Qwen SSRM and unstructured reasoning baselines (see Appendix F.5 for details). In summary, SSRM consume more tokens than the unstructured reasoning baselines on MedCalcV2 Tasks, while token usage is comparable on MATH500 and GPQA-D. One factor contributing to the increased usage is redundant argument and variable referencing, as shown in Figure 2. We leave the development of a more efficient referencing mechanism to future work.

## 5 CONCLUSION

We have presented methods for scalably testing whether an LLM adheres to a prescribed reasoning strategy on specific critical tasks. Our methods combine a Semi-Structured Reasoning Model (SSRM), which outputs reasoning steps in a semi-structured format, with methods for *auditing* these reasoning traces. We consider two challenging tasks: (a) extracting information from clinical text and (b) performing a series of calculations using the extracted values, based on either predefined rules or given formulas. These tasks are adapted from MedCalcBench, which has been cleaned, deduplicated, and restructured to separate the simpler formula-based tasks from the more complex rule-based ones.

We show that *structured reasoning audits* can reveal meaningful classes of likely reasoning errors for these tasks and qualitatively distinguish between the types of errors made across tasks and models. We further introduce *typicality audits*, which are probabilistic models trained on a corpus of semi-structured reasoning traces. Typicality audits approximate structured audits by (a) sorting predictions by typicality, (b) splitting predictions into quantiles, and (c) interpreting the top quantile as a pass and the bottom quantile as a fail. Both types of audits can be applied to few-shot prompted models.

Importantly, auditability appears to come without a cost in accuracy: overall, our Qwen SSRM model outperforms plausible baselines, including strong closed-source prompted models, an identically-trained unstructured baseline, and many other strong comparably-sized models. Likewise, the Llama SSRM demonstrates comparable performance relative to its identically-trained unstructured baseline.

[5] This also diverges from the MedCalcBench few-shot evaluation, which selects a single demonstration from the same calculator as the test instance.

## REPRODUCIBILITY STATEMENT

To facilitate reproducibility, we provide detailed information on the datasets used (please see Appendix F), implementation details (please see Appendix E), and code (Anonymous Github Link).

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

## A    LIMITATIONS

We demonstrate SSRMs' effectiveness on Qwen2.5-7B and Llama3.1-8B. Experiments with different architectures and larger scales could help clarify the generalizability of the technique.

While symbolic audits provide a novel mechanism for monitoring behavior of LLMs, they can only capture some aspects of intended behavior. If audit coverage is incomplete, a model might pass all audits while following a logically incorrect reasoning process. (This limitation is analogous to the use of unit tests in software development, where test coverage is often incomplete). Additionally, models can execute individual steps incorrectly—a failure mode that reasoning audits typically fail to detect.

Typicality audits identify reasoning traces that are unusual, which need not be correlated with traces that are incorrect (e.g., if a model has a high error rate, highly typical traces might still be incorrect.)

In this study, we conducted only preliminary experiments integrating test-time-scaling with audits. Further investigations into effectively combining audits with test-time-scaling methods—such as audit-based self-consistency—to show their utility during inference time are left for future work.

## B    BROADER IMPACTS

This paper introduces Semi-Structured Reasoning Models (SSRMs) and presents two types of audits to identify probable reasoning errors in the semi-structured reasoning traces: (1) hand-crafted or LLM-generated structured audits and (2) probabilistic model-based typicality audits. Our goal is to detect undesirable reasoning shortcuts for LLMs while maintaining good downstream performance.

## C    BACKGROUND: PROGRAM TRACE PROMPTING

Program Trace Prompting (PTP) Cohen & Cohen (2024) was proposed to make CoT explanations easier to analyze while preserving the generality and flexibility. In prior PTP work, existing few-shot CoT demonstrations were manually reformatted by wrapping them in a semi-formal syntax resembling a program trace. Functionally, the trace format (1) identifies and names steps, (2) defines the input/output behavior of steps, and (3) replaces every CoT explanation in a demonstration with a chain of formalized steps. The named steps were also documented with a Python "stub" that specifies type signatures for the inputs and outputs, and gives a short summary of the semantics of a step in a Python "docstring". Additionally, a top-level stub was created that specifies the task and contains, in its docstring, each of the sample traces. The resulting structure is referred to as a "partial program": it contains no executable code or pseudo-code, just documentation and a few high-level traces.

The partial program is then passed to an LLM along with a new program input, and the LLM is asked to predict a trace. An example of a partial program (with one demonstration, lightly edited for brevity) and the PTP system prompt is shown in Figure 5.

PTP performs comparably to traditional CoT prompting when CoT demonstrations are mapped directly to traces. A limitation of PTP, however, is that constructing the partial program requires the prompt designer to provide more explicit guidance on how to decompose a problem. SSRMs address this issue by using a fine-tuned model to generate partial programs as well as traces, thereby reducing the associated manual overhead.

## D    AUDIT-GUIDED QUALITATIVE ANALYSIS OF REASONING TRACES

### D.1    AUDITS IMPLEMENTATION

The output of SSRMs includes both the partial program and the trace, which appear as a series of function calls, as shown in Figure 6. These function calls may be nested. Before running the audits, each completed step is converted into a structured object that contains the following fields:

These function calls might be nested. Before audits are run, each completed step is converted to a structured object which always contains these fields.

**PTP partial program with one CoT demo encoded as a trace**

```python
def analyze_sentence(sentence: str) -> tuple[str, str, str]:
    """From a sentence about sports, extract the name of a player, an
    action, and an event.  The event will be an empty string if no event
    is mentioned in the sentence.
    """
    ...

def sport_for(x: str)-> str:
    """Return the name of the sport associated with a player, action, or event.
    """
    ...

def consistent_sports(sport1: str, sport2: str) -> bool:
    """Compare two descriptions of sports, and determine if they are consistent.

    Descriptions are consistent if they are the same, or if one is more
    general than the other.
    """
    ...

def sports_understanding(sentence):
    """Determine if a sentence about sports is plausible or not.

    >>> sports_understanding('Santi Cazorla scored a touchdown.')
    Calling analyze_sentence('Santi Cazorla scored a touchdown.')...
    ...analyze_sentence returned ('Santi Cazorla', 'scored a touchdown.', '')
    Calling sport_for('Santi Cazorla')...
    ...sport_for returned 'soccer'
    Calling sport_for('scored a touchdown.')...
    ...sport_for returned 'American football and rugby'
    Calling consistent_sports('soccer', 'American football and rugby')...
    ...consistent_sports returned False
    Final answer: no
    False
    """
    ...
```

**System Prompt Template for PTP**

```
Consider the program fragment below.  This program fragment is incomplete,
with key parts of the implementation hidden by replacing them
with "..." markers.

PROGRAM:
```python
{{PARTIAL_PROGRAM}}
```

QUESTION: Predict what the output of the program above will be, given
the input shown below.  Respond with the FULL program output, and ONLY
the expected program output: you will be PENALIZED if you introduce
any additional explanatory text.

>>> {{TASK_NAME}}({{TASK_INPUT}})
```

Figure 5: PTP partial program with one CoT demo encoded as traces (Top). System Prompt (Bottom).

**SSRMs Partial Program**

```
<partial_programs>

...omitted...

@traced
def analyze_input(input_str: str) -> tuple[str, list[str], list[str]]:
        """Accepts an input and extracts the question being asked, a list of rules to follow to answer
        ↪   the question, and the patient note.
        """
        ...

@traced
def get_data(formula: str, patient_note: str) -> list[str]:
        """Accepts a formula and a patient note, and extracts datapoints from the patient note required
        ↪   to evaluate the rule.
        """
        ...

...omitted...

</partial_programs>
```

**SSRMs output trace**

```
...omitted...

<program_trace>
   13        Calling get_data('Age: <50 years = 0 points, 50-59 years = +1 point, 60-69 years = +2 points,
   ↪  70-79 years = +3 points, >=80 years = +4 points', ['79-year-old gentleman'])...
   14          ...get_data returned '79 years old'
   15        Calling eval_rule('Age: <50 years = 0 points, 50-59
years = +1 point, 60-69 years = +2 points, 70-79 years = +3 points, >=80 years = +4 points', '79 years
↪  old')...
   16          ...eval_rule returned 3

...omitted...

</program_trace>
```

Figure 6: SSRMs partial program (Top). SSRMs output trace (Bottom).

**An Audit Example**

```
class MedCalcRulesAuditor(audit.Auditor):

...omitted...

    def test_each_rule_applied(self):
        df = self.df
          # there is one step that extracts the rules to apply
        _, rules, _ = df.query('step_fn == "analyze_input"').output.iloc[0]
        # check that there is a step to extract data for each rule
        get_data_steps = df[df.step_fn=='get_data']
        self.assertTrue(
            msg='one "get_data" step per rule',
            expr=(len(get_data_steps)==len(rules)))
        # check that there is a step to evaluate each rule

        eval_rule_steps = df[df.step_fn=='eval_rule']
        self.assertTrue(
            msg='one "eval_rule" step per rule',
            expr=(len(eval_rule_steps)==num_rules))
         # check that the first inputs of get_data are all rules,
        # and that every rule is used as input to get_data at least once
        self.assertTrue(
            msg='"get_data" called on all rules',
            expr=(set(rules) == set(get_data_steps.input1)))
```

Figure 7: An audit exmaple for MedCalcRules.

- `step_fn`: the name of the "function" being "traced", e.g., "`eval_rule`" for the second function call in the figure.

- `start_line`: the first line of the step, e.g., 15.

- `end_line`: the last line, e.g. 16. (If there are nested calls in between, the end line and start lines can be far apart).

- `str_inputs`: a string with the tuple of function inputs, e.g., "`('Age: <50 years = 0 points, 50-59 years = +1 point, 60-69 years = +2 points, 70-79 years = +3 points, >=80 years = +4 points', '79 years old')`"

- `str_output`: analogous, e.g., "3".

If the inputs can be parsed as a Python tuple, the following additional fields are added:

- `input`: a Python tuple of the inputs.

- `input1, input2, ...`: the Python values of the individual inputs.

- `len_input`: the length of the input tuple.

- `output`: the parsed Python value of the output.

- `output1, output2, ...`: when output is a tuple, the Python values of the individual outputs.

Finally, a Pandas DataFrame is constructed from all structured objects, with `NaN` used for missing fields (e.g., `input2` is absent for steps with only one input, and `output` is absent when `str_output` cannot be parsed as Python). The DSL for audits makes use of these DataFrames, combining DataFrame operations with a unit-test-like syntax. An example audit is provided in Figure 7.

### D.2 Example of a trace with reasoning flaws

To illustrate how audits can be useful, we randomly selected a problem from the MedCalcV2 Rules dataset (id #22) for which SSRM's output failed several audits. This problem asks the model to compute the Pneumonia Severity Index (PSI) for a 25-year-old male patient, given the patient note (which is about 150 words long) and 20 rules. (See Figure 8 (Top).)

In the resulting reasoning trace, only 19 rules of the 20 rules are called. For each of these 19 rules, an appropriate data extraction step is called and a result is returned, but the final score is computed by summing only 17 of the returned scores. Consequently, this trace fails r audits.[6]

- one `"get_data"` step per rule and one `"eval_rule"` step per rule both fail because neither a `get_data` nor `eval_rule` step was called for the 20th rule.

- `all outputs summed` fails because some rule outputs are not included in the sum for the final score.

- `get_data called on all rules` fails because the 20th rule was never used as an argument to `get_data`.

### D.3 Distinct patterns of audit failures indicate distinct reasoning flaws

The example in Figure 8 is typical of the MedCalcV2 Rules dataset: many audits are correlated, so examples that fail one often fail several others. In particular, the audits one `"get_data"` step per rule, one `"eval_rule"` step per rule, and `get_data called on all rules` often fail together. Table 7 reports the most common *combinations* of failed audits and summarizes the reasoning flaws typically associated with each combination (as determined by manual inspection). In brief, one recurring issue is the failure to check every rule, which we had noted in preliminary experiments with prompted models; but failure to accurately collect and sum all the results of rule evaluations are even more common for SSRMs.

---

[6]There is also at least one flaw in executing a *single* step, which we did not audit for this task: the (incorrectly) selected outputs are also incorrectly summed.

---

**Example of a flawed reasoning trace**

*Original input:*

```
A 25-year-old male patient presents at the clinic for a routine physical examination. The patient
↪  describes a recent episode of fever and altered mental state. He denies any past history of heart
↪  disease, renal disease, cerebrovascular disease, liver disease, or cancer. The patient is currently
↪  unemployed and lives alone. Upon physical examination, the patient has a measured body temperature of
↪  34 degrees celsisus, a heart rate of 101 beats per minute, and a respiratory rate of 32 breaths per
↪  minute. His blood pressure measures at 123/85 mm Hg. Laboratory findings show a hematocrit level of
↪  35, a blood glucose level of 15 mmol/L, and a BUN of 33.21 mg/dL. His sodium level is moderately low
↪  at 123 mmol/L. Arterial blood gas results show a PaO2 of 67 mm Hg and a slightly acidotic pH of 7.17.
↪  The patient's chest X-ray shows no signs of pleural effusion.

What is the patient's Pneumonia Severity Index (PSI)?

Calculate PSI Score: Pneumonia Severity Index for CAP by summing the results of the following rules:
    Age: Enter age in years (age score will be equal to age in years)
    Sex: Female = -10 points, Male = 0 points
    Nursing home resident: No = 0 points, Yes = +10 points
    Neoplastic disease: No = 0 points, Yes = +30 points

   ... rest of rules omitted ...
```

*Reasoning trace:*

```
Calling analyze_input( ...

rest of input omitted for brevity ...

...analyze_input returned (
  "What is the patient's Pneumonia Severity Index (PSI)?",
  ['Age: Enter age in years (age score will be equal to age in years)',
   'Sex: Female = -10 points, Male = 0 points',

... rest of rules omitted from return value ...

  ])
Calling get_data('Age: Enter age in years (age score will be equal to age in years)', ['25-year-old'])...
...get_data returned '25 years old'
Calling eval_rule('Age: Enter age in years (age score will be equal to age in years)', '25 years old')...
...eval_rule returned 25
Calling get_data('Sex: Female = -10 points, Male = 0 points', ['25-year-old male'])...
...get_data returned 'male'
Calling eval_rule('Sex: Female = -10 points, Male = 0 points', 'male')...
...eval_rule returned 0
Calling get_data('Nursing home resident: No = 0 points, Yes = +10 points', ['lives alone'])...
...get_data returned 'no'
Calling eval_rule('Nursing home resident: No = 0 points, Yes = +10 points', 'no')...
...eval_rule returned 0

... omitted ...

Calling get_data('Partial pressure of oxygen <60 mmHg or <8 kPa: No = 0 points, Yes = +10 points', ['PaO2
↪  of 67 mm Hg'])...
...get_data returned 'no'
Calling eval_rule('Partial pressure of oxygen <60 mmHg or <8 kPa: No = 0 points, Yes = +10 points',
↪  'no')...
...eval_rule returned 0
Calling sum_rules([25, 0, 0, 0, 0, 0, 0, 0, 20, 20, 0, 15, 0, 30, 20, 20, 0])...
...sum_rules returned 130
```

Figure 8: Example of a flawed reasoning trace.

| Set of failed audits | $N$ | $N_{\text{correct}}$ | Acc | Comments |
|---|---|---|---|---|
| (no failures) | 266 | 135 | 50.75% | |
| all outputs summed | 51 | 18 | 35.9% | Typically the score for 1–2 of the rules evaluated are not included in the final summation. |
| "get_data" called on all rules; one \"eval_rule" step per rule; one "get_data" step per rule | 26 | 9 | 34.6% | Typically one or more rules extracted from the input are not evaluated. |
| (all audits above fail) | 22 | 2 | 9.1% | Similar to the example of Section D.2. |

Table 7: More detailed analysis of the most common patterns of failed audits.

A more detailed qualitative analysis of the reasoning patterns reveals a more nuanced picture. Additional insight into unusual reasoning behaviors can be gained either by examining atypical combinations of audit failures or by instrumenting individual audits further.

As an example of the first type of analysis, only one trace (#291) fails exactly the two audits `all outputs summed` and one `"eval_rule" step per rule`. Manual inspection shows an unusual (but correct) reasoning pattern. For this example, the data are extracted for one particular rule is a common-separated list of three conditions relevant to the rule from the patient node. The model evaluates the rule three times on the same extraction, obtaining the correct total score for that rule. The final output is also correct. However, we argue that in a consequential task, detecting *anomalous reasoning patterns* is nearly as important as detecting errors, if the end goal is a reliable system with predictable behavior.

As an example of the second type of analysis, we instrumented the `all outputs summed` audit to report additional information. By tracking the total number of extracted rules, the number of rules scored, and the number of values summed, we observed that most of the time (more than 70%) only one or two rules were missed from the summation. In many of these cases, the omitted value was zero; thus, in more than 25% of the cases, the sum of the extracted values was numerically correct even though not all extracted values were included.

More interestingly, this instrumentation also revealed additional unusual reasoning patterns, in this case incorrect ones. In 7 of the failures for this audit, the number of values summed was *greater* than the number of rule evaluations. In most of these cases, the issue was again related to the problem of rules that match in multiple ways, as above: on these cases, the score reported for the rule is indicated by reporting a string containing the result of each match, as well as the final score, e.g., by returning "1 + 1 = 2" as the result of the rule evaluation.

## E  TRAINING DETAILS

Detailed hyperparameters configurations for both Stage 1 (SFT) and Stage 2 (RLVR) are provided in Table 8. We provide the detailed settings in subsequent subsections to support reproducibility.

Table 8: Hyperparameter settings for supervised fine-tuning (SFT) and reinforcement learning with verifiable rewards (RLVR). Both the semi-structured reasoning and CoT baseline settings use the same set of hyperparameters. †: max sequence length for SFT and max generation length for RLVR.

| Hyperparameter | SFT | RLVR |
|---|---|---|
| Optimizer | AdamW | AdamW |
| Actor Learning Rate | 1e-5 | 1e-6 |
| Weight Decay | 1e-4 | 0.1 |
| Warmup Ratio | 0.1 | 0.01 |
| Prompt Length | - | 2048 |
| Max Length$^{\dagger}$ | 16384 | 4096 |
| Loss Agg Mode | - | token_mean |
| Grad Clip | 0.2 | 1.0 |
| Batch Size | 128 | 256 |
| MiniBatch Size | - | 256 (On-Policy) |
| Num Responses Per Prompt | - | 8 |
| Temperature | - | 1.0 |
| Sequence Packing | False | True |
| Entropy Coeff | - | 0.0 |
| KL Loss Coeff | - | 0.0 |
| Epochs | 5 | 10 |

### E.1 SUPERVISED FINE-TUNING (SFT) DATA

We primarily follow the PTP approach (Cohen & Cohen, 2024) for generating semi-structured traces. A limitation of PTP, however, is that it requires manually written task-specific partial programs. For our experiments, we reuse the partial programs provided for BBH and manually construct those for GSM8K, MATH500, and MedCalcV2.

Beyond validating the final accuracy, we also perform a simple formatting check to remove samples whose partial programs or traces cannot be parsed. For the final dataset, we apply downsampling to balance the number of samples across tasks. Table 9 presents the distribution of SFT data for SSRMs.

Table 9: Distribution of Semi-Structured SFT data.

| Task | Count | % |
|------|-------|---|
| BBH | 2,727 | 64.76 |
| GSM8K | 393 | 9.33 |
| Math500 | 393 | 9.33 |
| MedCalc Formulas | 528 | 12.54 |
| MedCalc Rules | 170 | 4.04 |
| **Total** | **4,211** | **100.00** |

### E.2 SUPERVISED FINE-TUNING (SFT) CONFIGURATIONS

Figure 9 presents the system prompt template we used for SSRMs. The same system prompt is used for both Stage 1 and 2. Figure 10 shows a semi-structured reasoning trace from GSM8K used for SFT.

```
System Prompt Template for Semi-Structured Reasoning Models (SSRMs)

A conversation between User and Assistant. The User asks a question, and the Assistant solves it. The
↪  assistant first reasons through the problem by generating high-level partial programs with key parts
↪  hidden using "..." markers. It then simulates programs trace based on the incomplete partial programs.
↪  The partial program must be general enough to solve all instances of the problem type, not just
↪  specific examples. The partial programs and traces are enclosed within <partial_program>
↪  </partial_program> and <program_trace> </program_trace> tags, while the overall reasoning process and
↪  final answer are enclosed within <think> </think> and <answer> </answer> tags, respectively. You
↪  should also wrap your final answer in $\\boxed{{ANSWER}}$ if it is a mathematical expression.

Format:
<think>
<partial_program>
[Partial Program here]
</partial_program>
<program_trace>
[Program Trace here]
</program_trace>
</think>
<answer>
[Final Answer here]
</answer>
```

Figure 9: System Prompt Template for Semi-Structured Reasoning Models (SSRMs).

### E.3 REINFORCEMENT LEARNING WITH VERIFIABLE REWARDS (RLVR) CONFIGURATIONS

**Reward Design.** We employ two types of rule-based rewards functions for reinforcement learning:

- **Outcome Rewards:** The generated response will be assigned a reward of 1 for correct answers and 0 for incorrect answers. For the MedCalcV2 data, we follow the original proposed protocol and allow for a range-based evaluation; for the DAPO math data, we use an exact-match criterion.

**Semi-Structured Reasoning Trace for GSM8K**

```
<think>
<partial_program>
@traced
def analyze_input(input_str: str) -> tuple[str, ...]:
 """From an input, extract a tuple of individual questions and given values, as well as the final
 ↪   question to be answered.
 """
 ...

@traced
def convert_to_equations(input_questions: tuple[str, ...]) -> tuple[str, ...]:
 """Takes a list of questions and converts them to a list of mathematical equations.
 """
 ...

@traced
def simplify_equation(input_equation: str, given_values: tuple[str, ...]) -> str:
 """Takes an input equation and simplifies it by replacing a variable with the value of the variable as
 ↪   found in a list of given values if possible or performing algebraic operations if not, returning the
 ↪   simplified equation.
 """
 ...

</partial_program>

<program_trace>
Calling analyze_input("Two-fifths of the seats in an auditorium that holds 500 people are currently taken.
↪   It was found that 1/10 of the seats are broken. How many seats are still available?")...
...analyze_input returned ('An auditorium holds 500 people.', 'Two-fifths of the seats are currently
↪   taken.', '1/10 of the seats are broken.', 'How many seats are still available?')
Calling convert_to_equations(('An auditorium holds 500 people.', 'Two-fifths of the seats are currently
↪   taken.', '1/10 of the seats are broken.', 'How many seats are still available?'))...
...convert_to_equations returned ('total_seats = 500', 'occupied_seats = total_seats * 2/5',
↪   'broken_seats = total_seats * 1/10', 'available_seats = total_seats - occupied_seats - broken_seats')
Calling simplify_equation('available_seats = total_seats - occupied_seats - broken_seats', ('total_seats
↪   = 500', 'occupied_seats = total_seats * 2/5', 'broken_seats = total_seats * 1/10'))...
...simplify_equation returned 'available_seats = 500 - occupied_seats - broken_seats'
Calling simplify_equation('available_seats = 500 - occupied_seats - broken_seats', ('total_seats = 500',
↪   'occupied_seats = total_seats * 2/5', 'broken_seats = total_seats * 1/10'))...
...simplify_equation returned 'available_seats = 500 - (500 * 2/5) - broken_seats'
Calling simplify_equation('available_seats = 500 - (500 * 2/5) - broken_seats', ('total_seats = 500',
↪   'occupied_seats = total_seats * 2/5', 'broken_seats = total_seats * 1/10'))...
...simplify_equation returned 'available_seats = 500 - 200.0 - broken_seats'
Calling simplify_equation('available_seats = 500 - 200.0 - broken_seats', ('total_seats = 500',
↪   'occupied_seats = total_seats * 2/5', 'broken_seats = total_seats * 1/10'))...
...simplify_equation returned 'available_seats = 500 - 200.0 - (500 * 1/10)'
Calling simplify_equation('available_seats = 500 - 200.0 - (500 * 1/10)', ('total_seats = 500',
↪   'occupied_seats = total_seats * 2/5', 'broken_seats = total_seats * 1/10'))...
...simplify_equation returned 'available_seats = 500 - 200.0 - 50.0'
Calling simplify_equation('available_seats = 500 - 200.0 - 50.0', ('total_seats = 500', 'occupied_seats =
↪   total_seats * 2/5', 'broken_seats = total_seats * 1/10'))...
...simplify_equation returned 'available_seats = 250.0'
</program_trace>
</think>
<answer>
250
</answer>
```

Figure 10: Semi-Structured Reasoning Trace for GSM8K.

- **Format Rewards:** We require all models to format its responses using tags such as `<think>` and `<answer>`. For SSRMs specifically, additionally require the tags `<partial_program>` and `<program_trace>`, define at least three functions within the `<partial_program>` block, and exclusively invoke these functions within the `<program_trace>` block. Given the regular syntax of semi-structured reasoning traces, these constraints can be enforced via regular expressions.

# F  EXPERIMENTAL DETAILS

## F.1  MEDCALCBENCH V2

The original MedCalcBench (Khandekar et al., 2024) contains examples from 55 distinct *calculators*, including target quantities such as the SIRS score from Figure 1. In the original study, average scores were reported across all calculators: 37.9% in the zero-shot setting with GPT-4 and 50.9% in the one-shot setting. In the latter, the demonstration always used the same calculator as the test case, thereby evaluating the model's ability to extract data and reproduce an identical reasoning chain.

For our input, we concatenate the patient note and the original question, followed by a concise definition of the relevant formulas or rules. In the long-context CoT setting (for SSRMs), this concatenation serves as the sole input. In prompt-based settings, we employ a single two-shot CoT demonstration involving calculations of the same *type* (formula or rules), though not necessarily the same *calculator*, thereby testing the LLM's ability to extract data and perform a potentially different calculation. Therefore, MedCalcV2 scores are not directly comparable to those of MedCalcBench.

We implement two additional changes. First, we remove training samples in the original MedCalcBench that overlap with the test data to ensure a clean evaluation. Second, during testing, we discovered errors in results for the Glasgow Coma Scale Calculator: each ground-truth explanation duplicates the verbal-response rule and erroneously adds its value twice, leading to incorrect final scores. We manually correct these errors by deleting the duplicate lines and adjusting the final values in both the ground-truth explanations and the expected outputs. MedCalcV2 will be made available.

## F.2  TYPICALITY AUDIT CONFIGURATION

Results labeled HMM* are obtained via a grid search over hidden-state counts (1, 2, 5, 10) and n-gram sizes (1, 2, 3, 10, 25, 50), selecting the model with the lowest Bayesian Information Criterion (BIC) score (Dridi & Hadzagic, 2018). HMM are implemented using the `CategoricalHMM` class from `hmmlearn`, with preprocessing to convert sequences into n-gram representations. Each sequence is augmented with start and end tokens, an unknown-word token, and padded to a uniform length. We use the Fisher's exact test in `scipy.stats` for statistical significance of proportional differences.

## F.3  PROMPT FOR LLM-GENERATED AUDITS

Generated audits are created by prompting Claude-Sonnet-4-20250514 using the following prompt, replacing the label [TASKNAME] with the name of the task the audits are being generated for.

## F.4  ADDITIONAL RESULTS

Table 10 presents the comparison between the prompted Sonnet 3.5 model and a smaller prompted model, `Qwen2.5-7B-Instruct`, which is similar to the model we trained. The structured audits reveal that `Qwen2.5-7B-Instruct`'s performance diverges significantly from the larger Sonnet model. In the Formula task, Sonnet 3.5 exhibits no significant reasoning errors, whereas `Qwen2.5-7B-Instruct` frequently commits errors in the initial reasoning steps, resulting in substantially poorer outcomes. In the Rule task, `Qwen2.5-7B-Instruct` demonstrates a distinct failure mode than Sonnet model: it generates correctly structured solution traces, but then fails to execute each individual step correctly.

Table 11 shows results with LLM-generated audits on 21 tasks from the BBH benchmark suite. We report the number of lines of code in the generated audits, and the average number of audits that are run on each example. As a concise measure of the utility of the audits, we report the smallest $p$-value of any audit, as computed in Table 1 (i.e., for the null hypothesis that audit failure is not associated with incorrect outputs.) A small $p$-value indicates that some LLM-generated audit does

```
Prompt for LLM-Generated Audits

The attached file 'Example Audits' contains examples of audit functions which run on the traced outputs
↪   of functions called mocks. Each audit function tests the output to ensure that the mock has been run
↪   correctly by testing individual parts of the traced output, ensuring that each function the mock
↪   expects has been called, that the correct outputs lead to the correct inputs, and so on.

The attached file 'audit.py' contains the code which runs audit functions. Use this file to reference the
↪   expected structure of the dataframe that audit functions call on.

The attached file 'Audit Targets for [TASKNAME]' contains several traced outputs for a mock function ,
↪   [TASKNAME]. Generate a set of audit functions matching the format and construction of the examples
↪   from 'Example Audits', which will test other traced outputs of the function [TASKNAME]. Your
↪   generated audits should not programmatically generate the messages for success or failure.

Return only the python code for your output, with no extraneous introduction or afterward. Do not encase
↪   your output in backticks. Make sure to include imports and an if-main function.
```

Figure 11: Prompt for LLM-Generated Audits.

Table 10: Results of applying hand-coded structured audits to prompted models for MedCalcV2 tasks.

| | | | − accuracy and difference − | | | |
|---|---|---|---|---|---|---|
| | %Failed | Failing | Passing | Δ | $p$-val | description of audit |
| MedCalcV2 Formulas | 1.712 | 0.000 | 0.662 | 0.662 | 0.162 | one "get_data" step |
| | 2.055 | 0.000 | 0.664 | 0.664 | 0.086 | one "insert_variables" step |
| Claude 3.5 | 3.767 | 0.091 | 0.673 | 0.582 | 0.033 | "solve_formula" output is a number |
| (65.1% acc) | 9.247 | 0.593 | 0.657 | 0.064 | 0.870 | "solve_formula" output is formatted correctly |
| | 47.260 | 0.667 | 0.636 | -0.030 | 0.852 | "solve_formula" math is correct |
| | 3.425 | 0.000 | 0.296 | 0.296 | 0.126 | "solve_formula" output is a string |
| | 5.479 | 0.188 | 0.292 | 0.104 | 0.777 | "solve_formula" output is a number |
| | 7.192 | 0.381 | 0.279 | -0.102 | 0.487 | "solve_formula" output is formatted correctly |
| Qwen2.5-7B-Instruct | 29.110 | 0.376 | 0.249 | -0.128 | 0.140 | "solve_formula" math is correct |
| (28.6% acc) | 29.795 | 0.103 | 0.365 | 0.261 | 0.000 | one "get_data" step |
| | 30.137 | 0.102 | 0.366 | 0.264 | 0.000 | one "insert_variables" step |
| | 33.219 | 0.165 | 0.347 | 0.182 | 0.015 | one "analyze_input" step |
| MedCalcV2 Rules | 5.789 | 0.182 | 0.399 | 0.218 | 0.181 | analyze_input returns two values |
| | 14.737 | 0.196 | 0.420 | 0.223 | 0.028 | one step per rule with step_fn of "convert_units" |
| Claude 3.5 | 14.737 | 0.196 | 0.420 | 0.223 | 0.028 | one step per rule with step_fn of "get_data" |
| (38.7% acc) | 15.789 | 0.183 | 0.425 | 0.242 | 0.015 | one step per rule with step_fn of "check_rule" |
| | 17.105 | 0.169 | 0.432 | 0.263 | 0.005 | one step per rule with step_fn of "accumulate_score" |
| | 1.316 | 0.400 | 0.309 | -0.091 | 0.672 | one step per rule with step_fn of "get_data" |
| | 1.579 | 0.333 | 0.310 | -0.023 | 1.000 | one step per rule with step_fn of "convert_units" |
| Qwen2.5-7B-Instruct | 1.579 | 0.333 | 0.310 | -0.023 | 1.000 | one step per rule with step_fn of "accumulate_score" |
| (31.1% acc) | 1.579 | 0.333 | 0.310 | -0.023 | 1.000 | one step per rule with step_fn of "check_rule" |
| | 2.632 | 0.500 | 0.305 | -0.195 | 0.363 | one step with step_fn of "analyze_input" |
| | 4.737 | 0.389 | 0.307 | -0.082 | 0.630 | analyze_input returns two values |

indeed provide information about an "interesting" reasoning failure. Nearly half of the generated audits have $p$-values less than 0.05, including all four of the tasks with the highest error rates.

Table 12 shows results of generated structured audits on the same reasoning traces used in Table 4.

## F.5 TOKEN USAGE ANALYSIS

As shown in Table 13, SSRM consumes more tokens than the unstructured reasoning baselines on MedCalcV2 Rules and Formulas, whereas token usage is comparable on MATH500 and GPQA-Diamond. The higher token consumption primarily results from redundant arguments and variable referencing, as illustrated in Figure 2. Developing a more efficient variable referencing mechanism is left for future work.

| Dataset | Qwen Unstructured | Qwen SSRM |
|---|---|---|
| GSM8K | 319.78 | 841.72 |
| Math500 | 909.27 | 978.89 |
| MedCalcV2 Formulas | 411.80 | 1778.14 |
| MedCalcV2 Rules | 425.70 | 2260.87 |
| GPQA Diamond | 1608.33 | 1411.29 |
| MedQA | 359.25 | 1065.34 |

Table 13: Token usage of Qwen SSRM and corresponding unstructured baseline across datasets.

Table 11: Summary of LLM-generated audits on BBH tasks, using a prompted Claude Sonnet 3.5.

| Task | Task Acc | Avg Audits/Example | Code Lines | Min $p$-value |
|---|---|---|---|---|
| geometric shapes | 37.89% | 14.50 | 128 | $< 0.001$ |
| formal fallacies | 46.31% | 10.75 | 107 | $< 0.001$ |
| causal judgement | 57.48% | 11.75 | 88 | $< 0.001$ |
| dyck languages | 64.00% | 27.00 | 89 | $< 0.001$ |
| disambiguation qa | 82.63% | 10.98 | 93 | |
| ruin names | 83.16% | 10.00 | 105 | |
| penguins in a table | 87.21% | 11.01 | 114 | $< 0.05$ |
| multistep arithmetic two | 87.89% | 12.00 | 95 | |
| snarks | 91.53% | 85.72 | 109 | |
| date understanding | 87.89% | 11.33 | 88 | |
| logical deduction three objects | 87.89% | 10.99 | 94 | |
| movie recommendation | 91.05% | 13.98 | 90 | |
| reasoning about colored objects | 94.21% | 14.00 | 95 | |
| word sorting | 95.26% | 19.82 | 110 | $< 0.05$ |
| boolean expressions | 95.26% | 6.09 | 92 | $< 0.05$ |
| temporal sequences | 96.84% | 12.98 | 90 | |
| sports understanding | 97.37% | 7.00 | 71 | $< 0.05$ |
| hyperbaton | 97.89% | 7.00 | 69 | $< 0.001$ |
| tracking shuffled objects | 98.95% | 17.00 | 105 | $< 0.05$ |
| object counting | 100.00% | 9.00 | 70 | |
| web of lies | 100.00% | 15.00 | 94 | |

Table 12: LLM-generated structured audits Claude Sonnet 3.5 Prompted Models for MedCalcV2.

| | | − accuracy and difference − | | | | |
|---|---|---|---|---|---|---|
| | %Failed | Failing | Passing | Δ | $p$-val | description of audit |
| Formulas | 1.71 | 0.00% | 44.60% | 44.60% | * | one get_data step |
| | 0.34 | 0.00% | 43.99% | 43.99% | | one analyze_input step |
| | 0.68 | 0.00% | 44.14% | 44.14% | | analyze_input returns tuple with 2 elements |
| | 2.05 | 0.00% | 44.76% | 44.76% | * | one insert_variables step |
| | 4.79 | 35.71% | 44.24% | 8.53% | | convert_units called on each datapoint |
| | 3.42 | 50.00% | 43.62% | -6.38% | | convert_units' second input is a datapoint |
| | 0.68 | 50.00% | 43.79% | -6.21% | | convert_units's first input is the formula |
| | 0.68 | 50.00% | 43.79% | -6.21% | | insert_variables' first input is the formula |
| | 3.08 | 44.44% | 43.82% | -0.63% | | insert_variables' second input is an output of convert_units |
| | 2.74 | 37.50% | 44.01% | 6.51% | | get_data's inputs match the output of analyze_input |
| | 0.34 | 0.00% | 43.99% | 43.99% | | solve_formula's input is an output of insert_variables |
| | 92.81 | 42.80% | 57.14% | 14.34% | | final answer matches last solve_formula output |
| Rules | 5.79 | 18.18% | 39.94% | 21.76% | * | analyze_input returns tuple with 2 elements |
| | 14.74 | 19.64% | 41.98% | 22.33% | ** | get_data called for each rule |
| | 14.74 | 19.64% | 41.98% | 22.33% | ** | consistent rules across get_data steps |
| | 1.05 | 0.00% | 39.10% | 39.10% | | convert_units inputs are outputs of get_data |
| | 14.74 | 19.64% | 41.98% | 22.33% | ** | convert_units called for each rule |
| | 14.74 | 19.64% | 41.98% | 22.33% | ** | consistent rules across convert_units steps |
| | 2.63 | 0.00% | 39.73% | 39.73% | ** | check_rule inputs are outputs of convert_units |
| | 15.79 | 18.33% | 42.50% | 24.17% | ** | check_rule called for each rule |
| | 15.79 | 18.33% | 42.50% | 24.17% | ** | consistent rules across check_rule steps |
| | 0.79 | 0.00% | 38.99% | 38.99% | | accumulate_score inputs are outputs of check_rule |
| | 17.11 | 16.92% | 43.17% | 26.25% | ** | accumulate_score called for each rule |

## F.6 EVALUATION CONFIGURATIONS

We use `Lighteval` for all evaluations. For non-reasoning models, we report accuracy using greedy decoding. For reasoning models, we set the temperature to 0.6 and top-$p$ to 0.95. For the AIME24 dataset—where we observe high variance—we sample 32 responses using a temperature of 0.7 for non-reasoning models, while retaining the configurations for reasoning models, and report Pass@1.

Table 14: Comparison of Self-Consistency and Audit-Based Self-Consistency on MedcalcV2 Rule.

| Sampling Budget | Self-Consistency | Audit-Based Self-Consistency | Effective Samples |
|---|---|---|---|
| Greedy (Temp = 0) | 44.2 | 44.2 | - |
| Sampling (Temp = 0.7) | 44.2 | 44.2 | - |
| 3 | 46.3 | 45.3 | 306 (53.68%) |
| 5 | 45.3 | 46.8 | 522 (54.95%) |
| 7 | 45.3 | 45.3 | 764 (57.44%) |
| 9 | 45.3 | 45.3 | 1002 (58.60%) |
| 15 | 45.3 | 44.2 | 1702 (59.72%) |
| 30 | 45.2 | 46.3 | 3501 (61.42%) |
| 60 | 44.7 | 45.8 | 7071 (62.03%) |

### F.7 Test-Time-Scaling with Typicality Audits

To investigate the effectiveness of combining test-time-scaling with audits, we apply typicality audits (HMM*). We perform a grid search using the first half of the generated responses from the benchmark; to ensure data integrity, we evaluate the model only on the second half. We consider two variants here: vanilla self-consistency and audit-based self-consistency. Given a sampling budget of $k$ responses per question, in vanilla self-consistency we sample $k$ times per question and use majority voting to determine the final answer. In audit-based self-consistency, we divide the model-generated traces into tertiles: for traces in top tertile we perform no additional sampling, for those in the middle tertile we sample $k - 3$ additional times, and for those in the bottom tertile we sample $k - 1$ additional times. We report accuracy on the MedCalcV2 Rule tasks, along with the effective number of samples—i.e., the actual number generated under the audit-based procedure. For vanilla self-consistency, the total number of samples is $k \times n$, where $n$ is the number of questions in the corresponding benchmark.

As shown in Table 14, audit-based self-consistency consistently outperforms vanilla self-consistency given the same per-question sampling budget. More specifically, when $k = 5$, audit-based self-consistency outperforms vanilla self-consistency by 1.5 percentage points while using only 54.95% of the total sampling budget. These preliminary experiments demonstrate the effectiveness of combining typicality audits with test-time-scaling methods and suggest a promising direction for future research.

