# OpenReview forum: "Semi-structured LLM Reasoners Can Be Rigorously Audited"
_ICLR.cc/2026/Conference — Submitted to ICLR 2026_

### Official Review · Reviewer_oGJj · 2025-10-30

**Soundness:** 3
**Presentation:** 2
**Contribution:** 3
**Rating:** 4
**Confidence:** 3

**Summary:**

This paper addresses the critical problem of faithfulness and error detection in the reasoning of Large Language Models (LLMs) by proposing Semi-Structured Reasoning Models (SSRMs), which are trained to generate reasoning traces in a non-executable, Python-like syntax that explicitly names each step and its inputs/outputs for programmatic analysis. The core contribution is the introduction of three distinct auditing methods enabled by this structure: hand-crafted audits (DSL-based unit tests), LLM-generated audits (automating test creation), and typicality audits (a novel statistical approach flagging atypical reasoning step sequences). The authors provide strong empirical validation, primarily on a new MedCalcV2 benchmark, demonstrating that these audits effectively flag reasoning flaws that correlate with incorrect answers, and that training for this structured output (SSRM) does not compromise, but in fact improves overall task accuracy compared to identically-trained unstructured baselines.

**Strengths:**

1. The SSRM approach can avoid the rigidity of fully executable code (Program-of-Thought) but provides far more verifiability than unstructured free-text (Chain-of-Thought). The idea to parse the trace into a DataFrame and run unit-test-like audits is highly practical and interpretable.
2. The paper explores three different ways to perform audits, addressing rigor (hand-crafted), scalability (LLM-generated), and statistical novelty (typicality audits). The typicality audit, in particular, is a novel contribution for reasoning verification.

**Weaknesses:**

1. It is not clear to understand the correlation between %failed and Failing/Passing of the audits. Why sometimes passing is higher than failing with high delta, but we see %failed is also high?

2. The hand-crafted and LLM-generated audits seem highly effective for MedCalc, which is an inherently structured, rule-based task. It is less clear how this approach would scale to other reasoning tasks where the "correct" reasoning steps are not as clearly defined (in table 6). The "partial program" (the set of available functions) seems to require significant manual, task-specific design.

3. The paper mentions in Appendix E.5 and Table 11 that the SSRM traces are significantly longer than unstructured CoT traces (e.g., 4-5x more tokens for MedCalcV2). This implies a substantial increase in inference cost (time, memory, and computation) which is a major practical drawback, but this point is somewhat buried in the appendix.

**Questions:**

- Questions:
  - Page 6, Table 1 and 2: do the P values without * mean the result is not significant?
  - The manual effort to define the "partial program" (the vocabulary of reasoning functions like analyze_input, convert_units, evaluate_rule) seems like a potential bottleneck. How much effort was this for the MedCalcV2 task, and how do you see this step being managed for new, unseen domains? Do you have example how this can be generalized to non-MedCalcV2 task?
- Typos:
  - Page 6, Table 2: "sstep 2 output feeds into step 3 input" -> "step 2..." (double 's').

---

> ### Author Response · Authors · 2025-11-21
> **Response to Reviewer oGJj**
>
> Thank you very much for your valuable comments. We have revised the manuscript to incorporate your suggestions, and we will address your concerns in detail below:
>
> ---
>
> **Q1**: It is not clear to understand the correlation between %failed and Failing/Passing of the audits. Why sometimes passing is higher than failing with high delta, but we see %failed is also high?
>
> **A1**: Thank you for the question. In all cases, the passing rate is not statistically significantly higher than the failing rate. It is slightly higher for some audits that check for math errors, indicating that the audit is actually not very useful (This is probably because MedCalc's scoring allows numeric scores to be within a narrow range of the target value, so there is no penalty for rounding in numerical calculations, which also cause the audits to fail). We will note these in the final version.
>
> ---
>
> **Q2**: The hand-crafted and LLM-generated audits seem highly effective for MedCalc, which is an inherently structured, rule-based task. It is less clear how this approach would scale to other reasoning tasks where the "correct" reasoning steps are not as clearly defined (in table 6).
>
> **A2**: Thank you for your concerns. The experiments with BBH (Table 11) are intended to test LLM-generated audits on a broader set of less-structured tasks. The BBH problems include many that test different types of reasoning and less regular reasoning workflows.
>
>
> ---
>
> **Q3**: The paper mentions in Appendix E.5 and Table 11 that the SSRM traces are significantly longer than unstructured CoT traces (e.g., 4-5x more tokens for MedCalcV2). This implies a substantial increase in inference cost (time, memory, and computation) which is a major practical drawback, but this point is somewhat buried in the appendix.
>
> **A3**: Thank you for pointing this out. We also discuss this issue in the main text (lines 459 - 463). One factor contributing to the increases uage is redundant argument and variable refercing, as shown in Figure 2. Since this is not the main focus of this work, we leave the development of a more efficient referencing mechanisum to future work.
>
> ---
>
> **Q4**: Page 6, Table 1 and 2: do the P values without * mean the result is not significant?
>
> **A4**: Yes, that is correcy. For examplke, the various misuses of "convert units" in Table 2 happen, but the reasoning model recovers from those errors most of the time, so they do not lead to a higher error rate (as suggested by Figure 2, flawed reasoning need not affect the final outcome).
>
> The purpose of the experiments is to show that audits are analogous to unit tests: they can find reasoning errors but not all, and it is not expected that every audit will reveal problems frequently enough to be statistically verifiable.
>
> ---
>
> **Q5**: The manual effort to define the "partial program" (the vocabulary of reasoning functions like analyze_input, convert_units, evaluate_rule) seems like a potential bottleneck. How much effort was this for the MedCalcV2 task, and how do you see this step being managed for new, unseen domains? Do you have example how this can be generalized to non-MedCalcV2 task?
>
> **A5**: For the MedCalc tasks, we manually wrote partial programs during SFT data collection. This step can be time-consuming, but after training, **SSRMs can generate partial programs automatically**, along with the steps in the reasoning traces, and this is one of the paper's contributions. Table 6 in the main text further shows that SSRMs do not compromise performance even on out-of-domain tasks for which no manual partial programs were ever written.
>
> Yet, verifying the "quality" of the stubs is difficult and we view this as future work. In practice, it seems to be rare that stubs are unrelated to the task, but it is not clear that the "quality" of the stub can even be defined without knowing something of the desirable task workflow. For instance, it can be that models can get high accuracy with stubs that seem undesirable. For example, on some easier tasks they may produce very general stubs, like "answer_question", so the reaosning has little structure, yet accuracy can still remain high.
>
> ---
>
>
> **Q6**: Page 6, Table 2: "sstep 2 output feeds into step 3 input" -> "step 2..." (double 's').
>
> **A6**: Thank you for pointing out the typo. We have fixed it in the revised version.

---

> > ### Comment · Reviewer_oGJj · 2025-11-26
> >
> > Thank you for addressing my concerns. But I choose not to change my rating. I hope the paper can be further improved for a future publication. This work can be further strengthen if we can see clear improvements on most of reasoning tasks (in Table 6) comparing to unconstrained RL and understand how to obtain necessary audits highly related to final outputs for unseen tasks.

---

### Official Review · Reviewer_KMTn · 2025-10-31

**Soundness:** 2
**Presentation:** 2
**Contribution:** 3
**Rating:** 4
**Confidence:** 4

**Summary:**

This paper introduces Semi-Structured Reasoning Models (SSRMs), a method aimed at addressing the problem of faithfulness in LLM reasoning. The core idea is to train LLMs to produce reasoning traces in a semi-structured, non-executable Pythonic syntax. This structure, unlike free-form text, allows for automated audits to detect reasoning flaws. The authors propose and evaluate three auditing methods: hand-crafted audits, LLM-generated audits, and probabilistic typicality audits. They demonstrate that SSRMs can be effectively audited for errors while maintaining strong performance on several reasoning benchmarks compared to models of similar size.

**Strengths:**

- The paper tackles the highly significant and timely problem of ensuring the faithfulness of LLM reasoning. As LLMs are increasingly deployed in high-stakes domains, developing methods for auditing their reasoning processes is of critical importance, and this work makes a valuable contribution in this direction.
- The central idea of leveraging semi-structured representations for automated auditing is both novel and elegant. It offers a promising paradigm for moving beyond opaque, free-form text outputs towards more scrutable and verifiable reasoning chains.
- The authors provide compelling empirical evidence that their method does not compromise reasoning performance. Showing that the added structure for auditability comes at no cost to accuracy on a range of benchmarks is a strong and important result.

**Weaknesses:**

- The paper's clarity could be significantly improved, particularly concerning the methodological details. As it stands, some aspects of the implementation are challenging to fully understand, which may hinder reproducibility.
- The motivation behind choosing a "Pythonic syntax" would benefit from a more thorough discussion. Providing a comparison with alternative formalisms and explaining the trade-offs would help justify this specific design choice.
- The paper builds heavily on Program Trace Prompting (PTP) but would be more self-contained with a more detailed explanation of this prior work. This would help readers better appreciate the novel contributions of the current paper.
- The process for generating the semi-structured training data could be described in more detail. Clarifying how the reasoning traces are created and validated would strengthen the paper's claims.
- More information on the creation of function "stubs" would be helpful. Understanding whether these are pre-defined or model-generated, and how their quality is assured, is important for evaluating the auditing framework.
- The description of the audit creation process could be expanded. Elucidating how this process generalizes beyond the provided examples would bolster confidence in the method's broad applicability.

While the paper's central claim is compelling, the description of the reasoning generation and auditing process could be further developed. Answering some of the open questions about the framework—such as the generation of function stubs and training data—would substantiate the claims of rigorous auditability and strengthen the overall contribution.

**Questions:**

1.  Could you elaborate on the motivation for using a "Pythonic syntax"? The paper would be strengthened by a discussion of the alternatives that were considered and a clearer justification for this design choice.
2.  To improve the paper's self-containedness, could you provide a more detailed explanation of Program Trace Prompting (PTP) and clarify your specific modifications and extensions?
3.  Could you clarify the generation process for function stubs? It would be helpful to understand if they are manually defined or generated by an LLM, and how their quality and appropriateness for a given task are ensured.
4.  The paper would benefit from more details on the training data generation process. Could you elaborate on how semi-structured traces are created from existing datasets and what steps are taken to ensure their correctness?
5.  Could you expand on the audit creation process? Clarification on whether audits are generic or task-specific, and what the "minimal guidance" for LLM-generated audits entails, would be valuable for the reader.

---

> ### Author Response · Authors · 2025-11-21
> **Response to Reviewer KMTn (Part 1)**
>
> Thank you very much for your valuable comments. We have revised the manuscript to incorporate your suggestions, and we will address your concerns in detail below:
>
> ---
>
> **Q1**: The paper's clarity could be significantly improved, particularly concerning the methodological details. As it stands, some aspects of the implementation are challenging to fully understand, which may hinder reproducibility.
>
> **A1**: Thank you for the feedback. We have added a detailed description of the audit implementation in Appendix D.1 of the revised version, along with new analyses of flawed reasoning traces in Appendices D.2 and D.3. We also provide additional details on the construction of the SFT dataset and include the final data distribution in Table 9.
>
> ---
>
> **Q2**: The motivation behind choosing a "Pythonic syntax" would benefit from a more thorough discussion. Providing a comparison with alternative formalisms and explaining the trade-offs would help justify this specific design choice.
>
> **A2**: Thank you for the suggestions. The PTP [1] paper uses this format for their prompted model, and since we bootstrap both SFT and RL from a lightly reformatted PTP output, adopting the same format was a convenient and natural choice. For prompted models, this format is also effective: prior work such as PoT [2] shows that models can generate Python code easily, in a way that is compact and straightforward to parse. That said, other structured and parseable formats (e.g., JSON) would certainly be possible—they would simply require an additional parsing and restructuring step prior to fine-tuning.
>
> We also note that pretrained models like Qwen/Qwen-Math naturally tend to produce Python-based solutions for math problems, making this format especially compatible.
>
> [1] Watch Your Steps: Observable and Modular Chains of Thought, arxiv, 2024
>
> [2] Program of Thoughts Prompting: Disentangling Computation from Reasoning for Numerical Reasoning Tasks, TMLR 2023
>
> ---
>
> **Q3**: The paper builds heavily on Program Trace Prompting (PTP) but would be more self-contained with a more detailed explanation of this prior work. This would help readers better appreciate the novel contributions of the current paper.
>
> **A3**: Thank you for the suggestions, we have included a discussion in the Appendix C of the revised version.
>
> PTP [1] was proposed to make CoT explanations easier to analyze while preserving the generality and flexibility. In prior PTP work, existing few-shot CoT demonstrations were manually reformatted by wrapping them in a semi-formal syntax resembling a program trace. Functionally, the trace format (1) identifies and names steps, (2) defines the input/output behavior of steps, and (3) replaces every CoT explanation in a demonstration with a chain of formalized steps. The named steps were also documented with a Python "stub" that specifies type signatures for the inputs and outputs, and gives a short summary of the semantics of a step in a Python "docstring". Additionally, a top-level stub was created that specifies the task and contains, in its docstring, each of the sample traces. The resulting structure is referred to as a "partial program": it contains no executable code or pseudo-code, just documentation and a few high-level traces.
>
> The partial program is then passed to an LLM along with a new program input, and the LLM is asked to predict a trace.
>
> PTP performs comparably to traditional CoT prompting when CoT demonstrations are mapped directly to traces. A limitation of PTP, however, is that constructing the partial program requires the prompt designer to provide more explicit guidance on how to decompose a problem. SSRMs address this issue by using a fine-tuned model to generate partial programs as well as traces, thereby reducing the associated manual overhead.
>
> Finally, we note that reasoning audits, which are a central component of our pipeline, are not present in prior work and represent an additional contribution of our paper.
>
> [1] Watch Your Steps: Observable and Modular Chains of Thought, arxiv, 2024

---

> > ### Author Response · Authors · 2025-11-21
> > **Response to Reviewer KMTn (Part 2)**
> >
> > **Q4**: Could you clarify the generation process for function stubs? It would be helpful to understand if they are manually defined or generated by an LLM, and how their quality and appropriateness for a given task are ensured.
> >
> > **A4**: In the prior PTP [1] work, the stubs were pre-defined (as part of a task-specific prompt). In this work, SSRM, the stubs are **generated by the model**, along with the steps in the reasoning traces, so this is one of the paper's contributions.
> >
> > Defining the "quality" of the stubs is difficult and we view this as future work. In practice, it seems to be rare that stubs are unrelated to the task, but it is not clear that the "quality" of the stub can even be defined without knowing something of the desirable task workflow. For instance, it can be that models can get high accuracy with stubs that seem undesirable. For example, on some easier tasks they may produce very general stubs, like "answer_question", so the reaosning has little structure, yet accuracy can still remain high.
> >
> > [1] Watch Your Steps: Observable and Modular Chains of Thought, arxiv, 2024
> >
> > ---
> >
> > **Q5**: The process for generating the semi-structured training data could be described in more detail. Clarifying how the reasoning traces are created and validated would strengthen the paper's claims.
> >
> > **A5**: Thank you for the suggestion. In the revised version, we have added more description of the SFT data construction process, and Appendix Table 9 now reports the distribution of the final SFT dataset. Briefly, we follow the PTP [1] approach to prompt the model to generate semi-structured traces. In addition to the original BBH tasks, we manually wrote partial programs for MATH500, GSM8K, and MedCalcV2, and used these partial programs to prompt the model to produce traces for each task. Since all tasks have verifiable answers, we retain only traces whose final answers are correct. We also apply a simple format check to remove samples whose partial programs or traces cannot be parsed, and we perform downsampling to balance the number of samples across tasks. While some traces may contain flawed reasoning despite having correct final answers, the proposed reasoning audits provide a way to further validate such traces, which we plan to explore in future work.
> >
> > [1] Watch Your Steps: Observable and Modular Chains of Thought, arxiv, 2024
> >
> > ---
> >
> > **Q6**: Could you expand on the audit creation process? Clarification on whether audits are generic or task-specific, and what the "minimal guidance" for LLM-generated audits entails, would be valuable for the reader.
> >
> > **A6**: The audits in Table 1, 2, 4 were manually written by the authors after manually examining a few of the training examples, which is indeed time-consuming. The LLM-generated audits are generated using a prompt that contains 3 manually-written audits for a different task, and 1 correct reasoning traces from target task as minimal guidance. The full prompt can be found in Figure 8, Appendix. In Appendix D.1 of the revised version, we also include a detailed description of the structure and implementation of audits.

---

### Official Review · Reviewer_sE1b · 2025-10-31

**Soundness:** 1
**Presentation:** 2
**Contribution:** 2
**Rating:** 2
**Confidence:** 4

**Summary:**

This paper proposes to train semi-structured reasoning models (SSRMs), reasoning language models that output reasoning traces in a structured, Python-like format, whereas execution is done primarily via a language model itself. The main advantage is that the structure enables users to do "audits" of the reasoning process automatically -- applying rules that can verify certain properties of the reasoning process, rather than just the outcome. The authors propose a pipeline to train these models by inferring semi-structured traces from existing CoT traces. Experiments on domains ranging from bigbench-hard tasks, MATH and a medical domain show that SSRMs can be audited via hand-crafted and learned (typicality) rules.

**Strengths:**

The paper tackles a significant, well-motivated problem of how to test the validity of LLM reasoning beyond looking at their final answer. The general idea of proposing a loose structure makes sense, and it not being fully symbolic gives it flexibility to work across domains.

While this has broadly been explored, the idea of allowing users to write programmatic audits is novel as far as I'm aware.

**Weaknesses:**

The paper lacks important details about most of the method. Central to SSRMs is the format that is enforced, but the format itself is only vaguely described (there's the example in Figure 1, but the format is barely mentioned in Section 3). I'm confused by the fact that the representation is a Pandas DataFrame, since programs are hierarchical (and even if the trace is just linear, each function call has a variable number of arguments, which I would assume map to columns in the data frame). Thus, I don't really understand what is the structure that the paper is proposing.

Broadly, the idea of structuring chain-of-thought reasoning has been explored before (e.g., see Natural Programs [1], from NeurIPS '23). Thus, this is not really new. The new angle seems to be the programmatic audits. But understanding what kinds of audits are easy/hard to specify crucially requires seeing the format described in more detail. For instance, Table 1 mentions this "solve formula math is correct" audit -- obviously checking math in general is extremely hard, and the paper only mentions that it "uses Python’s eval function". Thus, I'm also unclear on how these audits are implemented, and what range of interesting errors do they indeed end up catching.

For the evaluation, the main results are about the audits themselves. The audits in Tables 1, 2 and 4 seem very ad hoc -- why these specifically? It seems like they were written by the authors themselves. It would be more reassuring if existing audits that we know people are already interested in could be demonstrated in your framework. Otherwise, it's hard to interpret the results. At the very least, the paper should show interesting examples of errors caught by these audits and why they are important to detect.

Thus, since the focus is on the ability to audit reasoning, I believe the paper should (1) describe the format in much more detail, (2) either find well-motivated audits from somewhere else, or show more evidence that the author's hand-crafted (or typicality) audits can capture interesting and important errors in LLM reasoning across the benchmarks.

[1] https://arxiv.org/pdf/2306.03872

**Questions:**

- Is the format flat, or is it hierarchical? If it's the latter, how does hierarchy get represented in the data frame?
- In Supervised Fine-Tuning (Section 3), the paper mentions that traces are extracted from existing datasets, but only "aces that yield correct final answer are retained". How is this check implemented, since you cannot just execute the traces using Python?
- What rate of structured traces were discarded by this rule?
- Does your full pipeline include SFT and GRPO? What fraction of compute is allocated to both? What's the delta from just doing SFT?
- Are there good examples of interesting errors your audits catch in MedCalcV2, where auditing seems to be the most important?

---

> ### Author Response · Authors · 2025-11-21
> **Response to Reviewer sE1b (Part 1)**
>
> Thank you very much for your valuable comments. We have revised the manuscript to incorporate your suggestions, and we will address your concerns in detail below:
>
> ---
>
> **Q1**: The paper lacks important details about most of the methods. Confused by the fact that the representation is a Pandas df. (Describe the format in much more detail)
>
> **A1**: Thank you for the question. We have provided a detailed implementation description in Appendix D.1 of the revised version.
>
> Briefly, the SSRM output includes the partial program enclosed in `<partial_program>` tags, e.g.,
> ```
> <partial_programs>
> (...omitted...)
> @traced
> def analyze_input(input_str: str) -> tuple[str, list[str], list[str]]:
> 	"""Accepts an input and extracts the question being asked, a list of rules to follow to answer the question, and the patient note.
> 	"""
> 	...
>
> @traced
> def get_data(formula: str, patient_note: str) -> list[str]:
> 	"""Accepts a formula and a patient note, and extracts datapoints from the patient note required to evaluate the rule.
> 	"""
> 	...
> (...omitted...)
> </partial_programs>
> ```
>
>
> And trace enclosed in `<program_trace>` tags, which looks like a series of function calls, e.g.,
> ```
> <program_trace>
> (...omitted...)
>    13	Calling get_data('Age: <50 years = 0 points, 50-59 years = +1 point, 60-69 years = +2 points, 70-79 years = +3 points, ≥80 years = +4 points', ['79-year-old gentleman'])...
>    14	...get_data returned '79 years old'
>    15	Calling eval_rule('Age: <50 years = 0 points, 50-59 years = +1 point, 60-69 years = +2 points, 70-79 years = +3 points, ≥80 years = +4 points', '79 years old')...
>    16	...eval_rule returned 3
> (...omitted...)
> </program_trace>
> ```
>
> These function calls might be nested.  Before audits are run, each completed step is converted to a structured object which always contains these fields
>   - `step_fn`: the name of the "function" being traced", e.g., `eval_rule` for the second function call above.
>   - `start_line`: the first line of the step, e.g., 15.
>   - `end_line`: the last line, e.g., 16 (If there are nested calls in between, the end line and start lines can be far apart).
>   - `str_inputs`: a string with the tuple of function inputs, e.g., "`('Age: <50 years = 0 points, 50-59 years = +1 point, 60-69 years = +2 points, 70-79 years = +3 points, ≥80 years = +4 points', '79 years old')`"
>   - `str_output`: analogous, e.g., "`3`".
>
> If the inputs can be parsed as a Python tuple, then the following additional fields are added:
>   - `input`: a Python tuple of the inputs.
>   - `input1, input2, ...`: the Python values of the individual inputs.
>   - `len_input`: the length of the input tuple.
>   - `output`: the parsed Python value of the output.
>   - `output1, output2, ...`: when output is a tuple, the Python values of the individual outputs.
>
> Finally, a Pandas DataFrame is constructed from all structured objects, with `NaN` being used for missing fields (e.g., `input2` would be misisng for `step_fn` with only one input, and `output` would be missing if the `output_str` could not be parsed as Python).
>
> The DSL for audits makes use of these dataframes, combining dataframe oeprations with a unit-test-like syntax. Here is an example:
>
> ```
> class MedCalcRulesAuditor(audit.Auditor):
>
>     (...omitted...)
>
>     def test_each_rule_applied(self):
>         df = self.df
> 	  # there is one step that extracts the rules to apply
>         _, rules, _ = df.query('step_fn == "analyze_input"').output.iloc[0]
>         # check that there is a step to extract data for each rule
>         get_data_steps = df[df.step_fn=='get_data']
>         self.assertTrue(
>             msg='one "get_data" step per rule',
>             expr=(len(get_data_steps)==len(rules)))
>         # check that there is a step to evaluate each rule
>
>         eval_rule_steps = df[df.step_fn=='eval_rule']
>         self.assertTrue(
>             msg='one "eval_rule" step per rule',
>             expr=(len(eval_rule_steps)==num_rules))
> 	  # check that the first inputs of get_data are all rules,
>         # and that every rule is used as input to get_data at least once
>         self.assertTrue(
>             msg='"get_data" called on all rules',
>             expr=(set(rules) == set(get_data_steps.input1)))
> ```

---

> > ### Author Response · Authors · 2025-11-21
> > **Response to Reviewer sE1b (Part 2)**
> >
> > **Q2**: The idea of structuring chain-of-thought reasoning has been explored before (e.g., see Natural Programs [1])
> >
> > **A2**: Thank you for pointing this out. This is indeed relevant, but there are also important differences. One difference is that [1] models proofs, not algorithms: they specify premises that are relied on, not inputs that are to be transformed. The advantage of [1] is that correctness of individual steps can be achieved through self-verification. Here we consider correctness of reasoning processes, not individual steps, using the idea of reasoning audits, which as noted are not used in prior work. (It may seem counterintuitive that all included steps can be correct but the reasoning process is incorrect, but the errors of omission in our Figure 1-2 examples illustrate how this is possible. Errors of this sort are also noted elsewhere, for example [2]).
> >
> > In [1], the authors also noted that incorrect chains can lead to correct solutions (and did not propose a solution). (At bottom of page 8).
> >
> >
> > [1] Deductive Verification of Chain-of-Thought Reasoning, NeurIPS, 2023
> >
> > [2] Language Models Are Greedy Reasoners: A Systematic Formal Analysis of Chain-of-Thought, ICLR, 2023
> >
> > ---
> >
> > **Q3**: The audits in Table 1, 2, and 4 seem very ad hoc, why these specifically? It would be more reassuring if existing audits that we know people are already interested in could be demonstrated in your framework.
> >
> > **A3**: Thank you for the question. The audits in Table 1, 2, and 4 were manually written by the authors after manually examining a few of the training examples. Since, the idea of reasoning audits is, to our best knowledge, novel, this means that there are no previously-existing datasets of audits from "somewhere else" to use as benchmark data. (and comparability would in any case require some sort of structured reasoning traces). We believe that it is an accepted methodology to introduce plausible new data to study when exploring novel phenomena, and that generating an initial corpus of plausible audits (manually and LLM-generated) is a contribution of the paper.
> >
> > ---
> >
> > **Q4**: In SFT, only "traces that yield correct final answer are retained". How is this check implemented, since you cannot just execute the traces using Python?
> >
> > **A4**: We only check the accuracy of the final answer. We demonstrate an example trace for GSM8K in Figure 6 of the Appendix. We extract the answer from answer tags and check its accuracy. Traces with correct final answers are retained. We also conduct a simple format check to filter out samples whose partial programs or traces cannot be parsed. For the final dataset, we additionally perform downsampling to balance the number of samples across tasks. We include this description in the revised version, along with the distribution of the SFT data in Appendix Table 8 (and in the next question).
> >
> > ---
> >
> > **Q5**: What rate of structured traces were discarded by this rule?
> >
> > **A5**: The discard rate are as follows: 26.20% for BBH, 6.26% for GSM8K, 63.82% for MATH500, and 74.26% for MedcalcV2. We also performed downsampling to maintain balance across tasks.
> > The final dataset sizes are
> >
> > | Dataset            | Final Count |
> > |--------------------|-------------|
> > | Total              | 4211        |
> > | BBH                | 2727        |
> > | GSM                | 393         |
> > | Math500            | 393         |
> > | MedCalc Formulas   | 528         |
> > | MedCalc Rules      | 170         |
> >
> > ---
> >
> > **Q6**: Does your full pipeline include SFT and GRPO? What fraction of compute is allocated to both? What’s the delta from just doing SFT?
> >
> > **A6**: Yes, our full training pipeline includes both SFT and GRPO. The relevant hyperparameters are provided in Appendix Table 7. In brief, we train SFT for 5 epochs on ~4K samples and GRPO for 10 epochs on ~19K samples. We also report the delta in Table 6 of the main text: the **+SFT** column reflects the performance of SFT only checkpoint, and the **++RL** column reflects the performance of the SFT + RL checkpoint.

---

> > > ### Author Response · Authors · 2025-11-21
> > > **Response to Reviewer sE1b (Part 3)**
> > >
> > > **Q7**: Are there good examples of interesting errors your audits catch in MedCalcV2, where auditing seems to be the most importan
> > >
> > > **A7**: Thank you for the question. We have included additional analysis in Appendix D.2 and D.3 of the revised version. We summarize it below for reference.
> > >
> > > We randomly selected a problem from the MedCalcV2 Rules dataset (id \#22) for which SSRM's output failed several audits.
> > >
> > > This problem asks the model to compute the Pneumonia Severity Index (PSI) for a 25-year-old male patient, given the patient note (which is about 150 words long) and 20 rules.
> > >
> > > ```
> > > (Original Input)
> > > A 25-year-old male patient presents at the clinic for a routine physical examination. The patient describes a recent episode of fever and altered mental state. He denies any past history of heart disease, renal disease, cerebrovascular disease, liver disease, or cancer. The patient is currently unemployed and lives alone. Upon physical examination, the patient has a measured body temperature of 34 degrees celsisus, a heart rate of 101 beats per minute, and a respiratory rate of 32 breaths per minute. His blood pressure measures at 123/85 mm Hg. Laboratory findings show a hematocrit level of 35, a blood glucose level of 15 mmol/L, and a BUN of 33.21 mg/dL. His sodium level is moderately low at 123 mmol/L. Arterial blood gas results show a PaO2 of 67 mm Hg and a slightly acidotic pH of 7.17. The patient's chest X-ray shows no signs of pleural effusion.
> > >
> > > What is the patient's Pneumonia Severity Index (PSI)?
> > >
> > > Calculate PSI Score: Pneumonia Severity Index for CAP by summing the results of the following rules:
> > >     Age: Enter age in years (age score will be equal to age in years)
> > >     Sex: Female = -10 points, Male = 0 points
> > >     Nursing home resident: No = 0 points, Yes = +10 points
> > >     Neoplastic disease: No = 0 points, Yes = +30 points
> > >     (...rest of rules omitted...)
> > >
> > > (Reasoning trace)
> > > Calling analyze_input( ...rest of input omitted for brevity...)
> > > ...analyze_input returned (
> > >   "What is the patient's Pneumonia Severity Index (PSI)?",
> > >   ['Age: Enter age in years (age score will be equal to age in years)',
> > >    'Sex: Female = -10 points, Male = 0 points',
> > >     (..rest of rules omitted from return value...)
> > >   ])
> > > Calling get_data('Age: Enter age in years (age score will be equal to age in years)', ['25-year-old'])...
> > > ...get_data returned '25 years old'
> > > Calling eval_rule('Age: Enter age in years (age score will be equal to age in years)', '25 years old')...
> > > ...eval_rule returned 25
> > > Calling get_data('Sex: Female = -10 points, Male = 0 points', ['25-year-old male'])...
> > > ...get_data returned 'male'
> > > Calling eval_rule('Sex: Female = -10 points, Male = 0 points', 'male')...
> > > ...eval_rule returned 0
> > > Calling get_data('Nursing home resident: No = 0 points, Yes = +10 points', ['lives alone'])...
> > > ...get_data returned 'no'
> > > Calling eval_rule('Nursing home resident: No = 0 points, Yes = +10 points', 'no')...
> > > ...eval_rule returned 0
> > > (...omitted...)
> > > Calling get_data('Partial pressure of oxygen <60 mmHg or <8 kPa: No = 0 points, Yes = +10 points', ['PaO2 of 67 mm Hg'])...
> > > ...get_data returned 'no'
> > > Calling eval_rule('Partial pressure of oxygen <60 mmHg or <8 kPa: No = 0 points, Yes = +10 points', 'no')...
> > > ...eval_rule returned 0
> > > Calling sum_rules([25, 0, 0, 0, 0, 0, 0, 0, 20, 20, 0, 15, 0, 30, 20, 20, 0])...
> > > ...sum_rules returned 130
> > > ```
> > >
> > > In the resulting reasoning trace, only 19 rules of the 20 rules are called. For each of these 19 rules, an appropriate data extraction step is called and a result is returned, but the final score is computed by summing only 17 of the returned scores.
> > > Consequently, this trace fails 4 audits (There is also at least one flaw in executing a *single* step, which we did not audit for this task: the (incorrectly) selected outputs are also incorrectly summed.)
> > >    - `"get_data" step per rule` and `one "eval_rule" step per rule` both fail because neither `get_data` nor `eval_rule` step was called for the 20th rule.
> > >    - `all outputs summed` fails because some rule outputs are not included in the sum for the final score.
> > >    - `get_data called on all rules` fails because the 20th rule was never used as an argument to `get_data`.

---

> > > > ### Author Response · Authors · 2025-11-21
> > > > **Response to Reviewer sE1b (Q7 cont.)**
> > > >
> > > > The above example is typical of the MedCalcV2 Rules dataset: many audits are correlated, so examples that fail one often fail several others. In particular, the audits `one "get_data" step per rule`, `one "eval_rule" step per rule`, and `get_data called on all rules` often fail together.
> > > > The following table reports the most common *combinations* of failed audits and summarizes the reasoning flaws typically associated with each combination (as determined by manual inspection).
> > > > In brief, one recurring issue is the failure to check every rule, which we had noted in preliminary experiments with prompted models; but failure to accurately collect and sum all the results of rule evaluations are even more common for SSRMs.
> > > >
> > > >
> > > > | Set of failed audits | N | N_correct | Acc | Comments |
> > > > |----------------------|---|-----------|------|----------|
> > > > | (no failures) | 266 | 135 | 50.75% | |
> > > > | `all outputs summed` | 51 | 18 | 35.3% | Typically the score for 1–2 of the rules evaluated are not included in the final summation. |
> > > > | `"get_data" called on all rules; one "eval_rule" step per rule; one "get_data" step per rule` | 26 | 9 | 34.6% | Typically one or more rules extracted from the input are not evaluated. |
> > > > | (all audits above fail) | 22 | 2 | 9.1% | Similar to the example of Section D.2. |
> > > >
> > > >
> > > > A more detailed qualitative analysis of the reasoning patterns reveals a more nuanced picture. Additional insight into unusual reasoning behaviors can be gained either by examining atypical combinations of audit failures or by instrumenting individual audits further.
> > > >
> > > > As an example of the first type of analysis, only one trace (\#291) fails exactly the two audits `all outputs summed` and `one "eval_rule" step per rule`. Manual inspection shows an unusual (but correct) reasoning pattern. For this example, the data are extracted for one particular rule is a common-separated list of three conditions relevant to the rule from the patient node. The model evaluates the rule three times on the same extraction, obtaining the correct total score for that rule. The final output is also correct. However, we argue that in a consequential task, detecting *anomalous reasoning patterns* is nearly as important as detecting errors, if the end goal is a reliable system with predictable behavior.
> > > >
> > > > As an example of the second type of analysis, we instrumented the `all outputs summed` audit to report additional information. By tracking the total number of extracted rules, the number of rules scored, and the number of values summed, we observed that most of the time (more than 70\%) only one or two rules were missed from the summation.
> > > > In many of these cases, the omitted value was zero; thus, in more than 25\% of the cases, the sum of the extracted values was numerically correct even though not all extracted values were included.
> > > >
> > > > More interestingly, this instrumentation also revealed additional unusual reasoning patterns, in this case incorrect ones. In 7 of the failures for this audit, the number of values summed was *greater* than the number of rule evaluations. In most of these cases, the issue was again related to the problem of rules that match in multiple ways, as above: on these cases, the score reported for the rule is indicated by reporting a string containing the result of each match, as well as the final score, e.g., by returning "1 + 1 = 2" as the result of the rule evaluation.

---

### Official Review · Reviewer_KRyG · 2025-10-31

**Soundness:** 3
**Presentation:** 2
**Contribution:** 3
**Rating:** 4
**Confidence:** 3

**Summary:**

The paper introduces semi structured reasoning as a way to audit the reasoning process of llms. The semi structured reasoning allows the authors to audit the reasoning in 3 ways: human audits, automated llm audits, and probabilistic audits based on reasoning patterns. The authors show how lllms trained to produce reasoning in this DSL can perform well with SFT and RL on medcalc bench while also producing auditable reasoning.

**Strengths:**

- The problem is clearly motivated and well presented.
- I particularly liked the typicality audits based on reasoning patterns
- The results on MedCalc are quite comprehensive

**Weaknesses:**

- My main concern: Why not generate certifiably correct reasoning? ****If the reasoning process follows a DSL, why not constrain generation to produce provably valid traces by construction (similar to Poesia et al.'s certified reasoning with LLMs)? The paper audits **after** generation, but doesn't explain why generation-time constraints aren't preferable. This would eliminate many errors rather than just detecting them.

Certified Deductive Reasoning with Language Models (https://arxiv.org/abs/2306.04031)
- There might be some circularity: The audits are generated with a stronger llm. How do we ensure that the audits themselves are correct?
- The results in table 6 are a little difficult to parse. It would be great if the table could split up into tables or figures to make different points more clear.
- 363 apply apply → apply both (?)
- **analysis of when audits fail:** The paper shows correlations between audit failures and errors but it would be great to see some more analysis:
    - FP rates
    - FN rates
    - What types of reasoning errors evade all three audit types

**Questions:**

Have you experimented with constrained generation to enforce valid DSL syntax during sampling?

---

> ### Author Response · Authors · 2025-11-21
> **Response to Reviewer KRyG (Part 1)**
>
> Thank you very much for your valuable comments. We have revised the manuscript to incorporate your suggestions, and we will address your concerns in detail below:
>
> ---
>
> **Q1**: Why not generate certifiably correct reasoning? Have you experimented with constrained generation to enforce valid DSL syntax during sampling?
>
> **A1**: Thank you for the question. Briefly, [1] produces proofes in a **structured formal language** (Peano), while we produce proofs in a semi-structured format which also allows for **informal** reasoning and plan execution. This means that our approach applies more broadly, beyond the math and logic tasks [1] considers, but reduces the benefit from constrained generation. Prior analysis shows that the Python-like syntax is very easy for models to learn: [2] Table 4 shows DSL syntax is almost always followed even in prompted models and even without constrained generation (>99% correct, and 99.7% correct ignoring issues around nesting quotations in long Python strings, which are not common on these problems).
>
> [1] Certified Deductive Reasoning with Language Models, TMLR 2024.
>
> [2] Watch Your Steps: Observable and Modular Chains of Thought, arxiv, 2024
>
> ---
>
> **Q2**: The audits are generated with a stronger llm. How do we ensure that the audits themselves are correct?
>
> **A2**: Thank you for the question. The audits are indeed generated with a stronger LLM, but the prompt includes 3 **hand-coded** audit examples from other tasks and 1 **correct reasoning trace** from the target task, which is informative and helps reduce failure modes. We agree that fully verifying audits is difficult without manual inspection, but in practice the vast majority of generated audits are simple data-flow assertions, which appear correct upon inspection. As an additional proxy for audit correctness, one could also check how many LLM-generated audits never fail on correctly solved examples, as audits are very likely to be correct.
>
> ---
>
> **Q3**: The results in table 6 are a little difficult to parse. It would be great if the table could split up into tables or figures to make different points more clear.
>
> **A3**: Thank you for the suggestions. We will make this clear in the final version. To better explain the current version: **unstr.** refers to the similarly trained CoT baseline, and **semi-str** refers to our SSRMs. **+SFT** refers to the SFT checkpoint, and **++RL** refers to the SFT + RL checkpoint. We also include comparably sized baseline models. Overall, Table 6 shows that semi-structured reasoning does not compromise performance.
>
> ---
>
> **Q4**: 363 apply apply
>
> **A4**: Thank you for pointing out the typo. We have fixed it in the revised version.

---

> ### Author Response · Authors · 2025-11-21
> **Response to Reviewer KRyG (Part 2)**
>
> **Q5**: It would be great to see some more analysis
>
> **A5**: Thank you for the suggestions. We have included additional analysis in Appendix D.2 and D.3 of the revised version. We summarize it below for reference.
>
> We randomly selected a problem from the MedCalcV2 Rules dataset (id \#22) for which SSRM's output failed several audits.
>
> This problem asks the model to compute the Pneumonia Severity Index (PSI) for a 25-year-old male patient, given the patient note (which is about 150 words long) and 20 rules.
>
> ```
> (Original Input)
> A 25-year-old male patient presents at the clinic for a routine physical examination. The patient describes a recent episode of fever and altered mental state. He denies any past history of heart disease, renal disease, cerebrovascular disease, liver disease, or cancer. The patient is currently unemployed and lives alone. Upon physical examination, the patient has a measured body temperature of 34 degrees celsisus, a heart rate of 101 beats per minute, and a respiratory rate of 32 breaths per minute. His blood pressure measures at 123/85 mm Hg. Laboratory findings show a hematocrit level of 35, a blood glucose level of 15 mmol/L, and a BUN of 33.21 mg/dL. His sodium level is moderately low at 123 mmol/L. Arterial blood gas results show a PaO2 of 67 mm Hg and a slightly acidotic pH of 7.17. The patient's chest X-ray shows no signs of pleural effusion.
>
> What is the patient's Pneumonia Severity Index (PSI)?
>
> Calculate PSI Score: Pneumonia Severity Index for CAP by summing the results of the following rules:
>     Age: Enter age in years (age score will be equal to age in years)
>     Sex: Female = -10 points, Male = 0 points
>     Nursing home resident: No = 0 points, Yes = +10 points
>     Neoplastic disease: No = 0 points, Yes = +30 points
>     (...rest of rules omitted...)
>
> (Reasoning trace)
> Calling analyze_input( ...rest of input omitted for brevity...)
> ...analyze_input returned (
>   "What is the patient's Pneumonia Severity Index (PSI)?",
>   ['Age: Enter age in years (age score will be equal to age in years)',
>    'Sex: Female = -10 points, Male = 0 points',
>     (..rest of rules omitted from return value...)
>   ])
> Calling get_data('Age: Enter age in years (age score will be equal to age in years)', ['25-year-old'])...
> ...get_data returned '25 years old'
> Calling eval_rule('Age: Enter age in years (age score will be equal to age in years)', '25 years old')...
> ...eval_rule returned 25
> Calling get_data('Sex: Female = -10 points, Male = 0 points', ['25-year-old male'])...
> ...get_data returned 'male'
> Calling eval_rule('Sex: Female = -10 points, Male = 0 points', 'male')...
> ...eval_rule returned 0
> Calling get_data('Nursing home resident: No = 0 points, Yes = +10 points', ['lives alone'])...
> ...get_data returned 'no'
> Calling eval_rule('Nursing home resident: No = 0 points, Yes = +10 points', 'no')...
> ...eval_rule returned 0
> (...omitted...)
> Calling get_data('Partial pressure of oxygen <60 mmHg or <8 kPa: No = 0 points, Yes = +10 points', ['PaO2 of 67 mm Hg'])...
> ...get_data returned 'no'
> Calling eval_rule('Partial pressure of oxygen <60 mmHg or <8 kPa: No = 0 points, Yes = +10 points', 'no')...
> ...eval_rule returned 0
> Calling sum_rules([25, 0, 0, 0, 0, 0, 0, 0, 20, 20, 0, 15, 0, 30, 20, 20, 0])...
> ...sum_rules returned 130
> ```
>
> In the resulting reasoning trace, only 19 rules of the 20 rules are called. For each of these 19 rules, an appropriate data extraction step is called and a result is returned, but the final score is computed by summing only 17 of the returned scores.
> Consequently, this trace fails 4 audits (There is also at least one flaw in executing a *single* step, which we did not audit for this task: the (incorrectly) selected outputs are also incorrectly summed.)
>    - `"get_data" step per rule` and `one "eval_rule" step per rule` both fail because neither `get_data` nor `eval_rule` step was called for the 20th rule.
>    - `all outputs summed` fails because some rule outputs are not included in the sum for the final score.
>    - `get_data called on all rules` fails because the 20th rule was never used as an argument to `get_data`.

---

> ### Author Response · Authors · 2025-11-21
> **Response to Reviewer KRyG (Q5 cont.)**
>
> The above example is typical of the MedCalcV2 Rules dataset: many audits are correlated, so examples that fail one often fail several others. In particular, the audits `one "get_data" step per rule`, `one "eval_rule" step per rule`, and `get_data called on all rules` often fail together.
> The following table reports the most common *combinations* of failed audits and summarizes the reasoning flaws typically associated with each combination (as determined by manual inspection).
> In brief, one recurring issue is the failure to check every rule, which we had noted in preliminary experiments with prompted models; but failure to accurately collect and sum all the results of rule evaluations are even more common for SSRMs.
>
>
> | Set of failed audits | N | N_correct | Acc | Comments |
> |----------------------|---|-----------|------|----------|
> | (no failures) | 266 | 135 | 50.75% | |
> | `all outputs summed` | 51 | 18 | 35.3% | Typically the score for 1–2 of the rules evaluated are not included in the final summation. |
> | `"get_data" called on all rules; one "eval_rule" step per rule; one "get_data" step per rule` | 26 | 9 | 34.6% | Typically one or more rules extracted from the input are not evaluated. |
> | (all audits above fail) | 22 | 2 | 9.1% | Similar to the example of Section D.2. |
>
> ---
>
> A more detailed qualitative analysis of the reasoning patterns reveals a more nuanced picture. Additional insight into unusual reasoning behaviors can be gained either by examining atypical combinations of audit failures or by instrumenting individual audits further.
>
> As an example of the first type of analysis, only one trace (\#291) fails exactly the two audits `all outputs summed` and `one "eval_rule" step per rule`. Manual inspection shows an unusual (but correct) reasoning pattern. For this example, the data are extracted for one particular rule is a common-separated list of three conditions relevant to the rule from the patient node. The model evaluates the rule three times on the same extraction, obtaining the correct total score for that rule. The final output is also correct. However, we argue that in a consequential task, detecting *anomalous reasoning patterns* is nearly as important as detecting errors, if the end goal is a reliable system with predictable behavior.
>
> As an example of the second type of analysis, we instrumented the `all outputs summed` audit to report additional information. By tracking the total number of extracted rules, the number of rules scored, and the number of values summed, we observed that most of the time (more than 70\%) only one or two rules were missed from the summation.
> In many of these cases, the omitted value was zero; thus, in more than 25\% of the cases, the sum of the extracted values was numerically correct even though not all extracted values were included.
>
> More interestingly, this instrumentation also revealed additional unusual reasoning patterns, in this case incorrect ones. In 7 of the failures for this audit, the number of values summed was *greater* than the number of rule evaluations. In most of these cases, the issue was again related to the problem of rules that match in multiple ways, as above: on these cases, the score reported for the rule is indicated by reporting a string containing the result of each match, as well as the final score, e.g., by returning "1 + 1 = 2" as the result of the rule evaluation.

---

### Author Response · Authors · 2025-11-21
**Summary of the revised version**

We thank the reviewers [R1(KRyG), R2(sE1b), R3(KMTn), R4(oGJj)] for their constructive feedback. We are glad that reviewers found the problem significant and our focus on auditability well-motivated [R1, R2, R3], and that the ideas of structured audits and typicality audits were viewed as interesting and novel [R1, R2, R3, R4].

Based on the feedback, we have revised our manuscripts accordingly and uploaded a new version for review. All changes are highlighted in blue. The key updates are summarized below:
   - We update the description of SFT data construction and include a distribution table in Appendix Table 8 [R2].
   - We include a detailed discussion of PTP and clarify differences and contributions in Appendix C [R3].
   - We add a detailed explanation of the audits structure and implementation in Appendix D.1. [R2, R3]
   -  We provide a detailed analysis of errors caught by audits in Appendix D.2, D.3. [R1, R2]

---

### Meta-Review · Area_Chair_oJJV · 2026-01-07

**Summary:**

The paper proposes Semi-Structured Reasoning Models (SSRMs), which generate Python-like reasoning traces so that the reasoning can be checked by audits. The audits include hand-written rules, audits generated by a stronger LLM, and a probabilistic “typicality” audit based on reasoning patterns. Reviewers agree the goal (auditing reasoning, not just answers) is important, and the typicality audit is a promising idea. However, key parts of the approach are not clearly specified, the novelty over prior “structured reasoning / natural programs / certified reasoning” lines is not fully established, and the evaluation leaves open questions about generality, reliability, and cost.

Pros
1. Clear motivation: improving trust and catching reasoning errors beyond final-answer accuracy
2.  Auditing framework is practical; typicality audits are an interesting addition

Cons
1. Method details are still hard to pin down: the exact trace format, how it maps to a DataFrame, how nesting/arguments are represented, and how audits are implemented and validated are unclear from the main paper
2. Limited novelty / missed comparison to constrained or certified generation: if a DSL is used, it is not well justified why audits are done after-the-fact instead of enforcing validity during generation, and the relationship to prior structured reasoning work is not clearly positioned
3. Audit reliability is uncertain: LLM-generated audits depend on a stronger model, raising concerns about circularity and correctness of the audits themselves
4. Generality is not convincingly demonstrated: audits work best on highly structured tasks like MedCalc; it is unclear how much manual “partial program” design is required and how well this transfers to less-structured reasoning tasks

**Reviewer Concerns:**

1. Clarifying the trace format and DataFrame representation: The rebuttal provides a more concrete description of how traces are parsed into structured fields and how audits operate over the resulting table, which helps resolve the basic confusion about “what is the structure.”
2. Example-driven explanation of audit failures: The added MedCalc case study and the audit-failure combination table provide useful intuition for what audits catch (especially errors of omission such as missing rules or missing terms in a sum).
3. Some presentation issues: The rebuttal acknowledges typos and promises to reorganize confusing tables, which likely improves readability.

**Reviewer Scores:**

N/A

---

### Decision · Program_Chairs · 2026-01-26

Reject